# Covalent disruptor of YAP-TEAD association suppresses defective Hippo signaling

Mengyang Fan[1,2,3†], Wenchao Lu[1,2,4†], Jianwei Che[1,2], Nicholas P Kwiatkowski[1,2], Yang Gao[1,2], Hyuk-Soo Seo[1,2], Scott B Ficarro[1,5], Prafulla C Gokhale[6], Yao Liu[1,2], Ezekiel A Geffken[1], Jimit Lakhani[1], Kijun Song[1], Miljan Kuljanin[7,8], Wenzhi Ji[1,2,4], Jie Jiang[1,2], Zhixiang He[1,2], Jason Tse[4], Andrew S Boghossian[9], Matthew G Rees[9], Melissa M Ronan[9], Jennifer A Roth[9], Joseph D Mancias[8], Jarrod A Marto[1,5], Sirano Dhe-Paganon[1,2], Tinghu Zhang[1,2,4]*, Nathanael S Gray[1,2,4]*

[1]Department of Cancer Biology, Dana-Farber Cancer Institute, Harvard Medical School, Boston, United States; [2]Department of Biological Chemistry and Molecular Pharmacology, Harvard Medical School, Boston, United States; [3]Hangzhou Institute of Medicine (HIM), Chinese Academy of Sciences, Hangzhou, China; [4]Department of Chemical and Systems Biology, ChEM-H, Stanford Cancer Institute, School of Medicine, Stanford University, Stanford, United States; [5]Blais Proteomics Center, Dana-Farber Cancer Institute, Harvard Medical School, Boston, United States; [6]Experimental Therapeutics Core, Dana-Farber Cancer Institute, Boston, United States; [7]Department of Cell Biology, Harvard Medical School, Boston, United States; [8]Division of Radiation and Genome Stability, Department of Radiation Oncology, Dana- Farber Cancer Institute, Harvard Medical School, Boston, United States; [9]Broad Institute of MIT and Harvard, Cambridge, United States

*For correspondence:
ztinghu8@stanford.edu (TZ);
nsgray01@stanford.edu (NSG)

†These authors contributed
equally to this work

Competing interest: See page
32

Reviewing Editor: Duojia Pan,
UT Southwestern Medical Center
and HHMI, United States

**Abstract** The transcription factor TEAD, together with its coactivator YAP/TAZ, is a key transcriptional modulator of the Hippo pathway. Activation of TEAD transcription by YAP has been implicated in a number of malignancies, and this complex represents a promising target for drug discovery. However, both YAP and its extensive binding interfaces to TEAD have been difficult to address using small molecules, mainly due to a lack of druggable pockets. TEAD is posttranslationally modified by palmitoylation that targets a conserved cysteine at a central pocket, which provides an opportunity to develop cysteine-directed covalent small molecules for TEAD inhibition. Here, we employed covalent fragment screening approach followed by structure-based design to develop an irreversible TEAD inhibitor MYF-03–69. Using a range of in vitro and cell-based assays we demonstrated that through a covalent binding with TEAD palmitate pocket, MYF-03–69 disrupts YAP-TEAD association, suppresses TEAD transcriptional activity and inhibits cell growth of Hippo signaling defective malignant pleural mesothelioma (MPM). Further, a cell viability screening with a panel of 903 cancer cell lines indicated a high correlation between TEAD-YAP dependency and the sensitivity to MYF-03–69. Transcription profiling identified the upregulation of proapoptotic *BMF* gene in cancer cells that are sensitive to TEAD inhibition. Further optimization of MYF-03–69 led to an in vivo compatible compound MYF-03–176, which shows strong antitumor efficacy in MPM mouse xenograft model via oral administration. Taken together, we disclosed a story of the development of covalent TEAD inhibitors and its high therapeutic potential for clinic treatment for the cancers that are driven by TEAD-YAP alteration.

## Editor's evaluation

The Hippo signaling pathway has emerged as a key signaling pathway in cancer and many other diseases, but there is a lack of high-quality chemical tools that would enable functional studies. Fan et al. disclose the development of covalent TEAD inhibitors and report on the therapeutic potential of this class of agents in the treatment of TEAD-YAP-driven cancers. The authors' claims and conclusions are well supported by the data. The optimized derivative represents a clear improvement from previously reported compounds and provides a high-quality probe for future studies.

## Introduction

The Hippo pathway is a highly conserved signaling pathway that regulates embryonic development, organ size, cell proliferation, tissue regeneration, homeostasis and is responsible for the development and progression of many malignancies *Ota and Sasaki, 2008*. Critical components of Hippo signaling pathway are TEAD transcription factors, that are present as a family of four highly homologous members (TEAD1-4) in mammals. Importantly, TEADs alone show minimal transcriptional activity and require binding with a coactivator YAP/TAZ to initiate efficient gene expression *Zhao et al., 2008*. YAP has been well characterized as an oncoprotein *Zanconato et al., 2016* and its tumorigenesis role is mostly associated with TEAD mediated YAP-dependent gene expression *Stein et al., 2015*. Tissue-specific deletion of the upstream regulators or overexpression of YAP itself results in hyperplasia, organ overgrowth and tumorigenesis *Zhou et al., 2009*. Tumor suppressor Merlin, encoded by *NF2* gene, and kinase LATS1/2 are known upstream regulators that cooperatively promote phosphorylation of YAP and hence induce its retention and degradation in the cytoplasm *Petrilli and Fernández-Valle, 2016*. However, loss of function mutations in *NF2* or *LATS1/2*, which occurs in >70% mesothelioma, promote YAP nuclear entry and binding with TEAD to drive oncogenesis *Bueno et al., 2016*. In addition to mesothelioma, high level of nuclear YAP has been associated with poor prognosis in non-small cell lung cancer (NSCLC), pancreatic cancer, and colorectal cancer (CRC)[3a]. Moreover, there is evidence to suggest that activation of the YAP-TEAD transcriptional program can be involved with drug resistance *Nguyen and Yi, 2019*. For example, *ALK*-rearranged lung cancer cells were shown to survive the treatment of ALK inhibitor alectinib through YAP1 activation *Tsuji et al., 2020*. In EGFR-mutant lung cancers, YAP-TEAD was observed to promote cell survival and induce a dormant state upon EGFR signaling blockade *Kurppa et al., 2020*. Taken together, targeting YAP-TEAD has emerged as a promising therapeutic strategy for cancer treatment.

Over the decades, efforts to directly target YAP/TAZ-TEAD have focused on either designing peptide-mimicking agents that bind in the interface between YAP and TEAD and disrupt this interaction or using phenotypic screening to identify small molecules that inhibit YAP-dependent signaling *Santucci et al., 2015*. For example, a cyclic YAP-like peptide has been developed as a potent disruptor of YAP-TEAD complex in cell lysates but failed to exert effects in cells due to poor cellular permeability *Zhang et al., 2014*. Recently, TEAD stability and the association with YAP protein was shown to be regulated by *S*-palmitoylation, a post-translational modification on a conserved cysteine located within the palmitate binding pocket (PBP) on YAP binding domain (YBD) of TEAD *Noland et al., 2016*. This finding led to the discovery of several small molecules: flufenamic acid (FA) *Pobbati et al., 2015*, MGH-CP1 *Li et al., 2020*, compound 2 *Holden et al., 2020* and VT101~107 *Tang et al., 2021*, that reversibly bind TEAD PBP. Additionally, inspired by thioester formation of the conserved cysteine with palmitate, covalent TEAD inhibitors TED-347 *Bum-Erdene et al., 2019*, DC-TEADin02 *Lu et al., 2019* and K-975 (*Kaneda et al., 2020*; *Figure 1—figure supplement 1*) have also been designed. TED-347 and DC-TEADin02 covalently bind with TEAD using a highly electrophilic chloromethyl ketone and vinyl sulfonamide warhead. In comparison, a less electrophilic arylamide warhead is used in the design of K-975, which shows a potent cellular and in vivo activity in mouse tumor models albeit employing a high dose. Although these compounds have adequately demonstrated that the palmitoylated cysteine is targetable, none of them are optimized compounds and the structure-activity study for each of these scaffolds has not been disclosed. We have previously reported MYF-01–37 which possesses a less electrophilic warhead compared to K-975 as a covalent TEAD inhibitor and demonstrated its utility to blunt the transcriptional adaptation in mutant EGFR-dependent NSCLC cells *Kurppa et al., 2020*. However, MYF-01–37 is a sub-optimal chemical probe which requires micromolar concentrations in cellular assays and displays poor pharmacokinetic properties.

Development of covalent chemical probes and drugs has experienced a revival in recent years resulting in a deluge of new modality targeting a range of cancer-relevant targets, such as KRAS$^{G12C}$, EGFR, BTK, JAK3 and others *Zhang et al., 2019*. The conventional approach to develop a covalent inhibitor is using a structure-guided design to equip a pre-existing reversible binder with an electrophile to target a nucleophilic residue on the target protein, most typically a cysteine. Alternatively, a recently developed approach involves screening a library of low molecular weight electrophilic fragments followed by medicinal chemical optimization *Resnick et al., 2019*. Here, we used such screening approach to identify the lead covalent fragment MYF-01–37. Subsequent structure-based design to engage a side pocket resulted in development of a more potent covalent TEAD inhibitor MYF-03–69, which inhibits the palmitoylation process of all four TEAD paralogs in vitro by occupying the Y-shaped pocket, disassociates the endogenous YAP-TEAD complex, downregulates YAP-dependent target gene expression, and preferentially impairs the proliferation of mesothelioma cells that exhibit defects in Hippo signaling. A comprehensive screen against 903 cancer cell lines *Yu et al., 2016* identified more YAP-TEAD dependent cells as sensitive to MYF-03–69 in addition to mesothelioma. Transcription profiling suggests that the upregulation of Bcl-2 modifying factor (*BMF*) gene correlated with antiproliferative response to TEAD inhibition. Further, in vivo pharmacological effect was achieved with orally bioavailable compound MYF-03–176, indicating a promising lead for therapy of cancers driven by Hippo signaling dysfunction.

## Results

### Covalent fragment screening identified MYF-01-37 as a covalent TEAD binder

To screen for a covalent lead compound for TEAD, we first assembled a biased covalent fragment library that was developed based on the flufenamic acid template and a cysteine-reactive acrylamide warhead (*Ábrányi-Balogh et al., 2018 Figure 1a*, *Figure 1—figure supplement 2*). We then conducted a medium-throughput screen of this library using mass spectrometry (MS) to detect protein-fragment adduct formation with recombinant, purified TEAD2 YBD protein. MYF-01–37 was identified as the fragment capable of efficient labeling while the majority of the fragment library failed to label the protein (*Figure 1—figure supplement 2*). Next, the protein labeled with MYF-01–37 was proteolytically digested and the resultant peptides were analyzed using tandem mass spectrometry (MS/MS) which confirmed labeling of C380, which is the cysteine previously reported to be *S*-palmitoylated in TEAD2 (*Figure 1a*).

### Structure-based optimization yields MYF-03-69, an irreversible inhibitor of TEAD palmitoylation

Next, we employed a structure-based fragment growing strategy to elaborate MYF-01–37 to develop a more potent and selective inhibitor (*Figure 1a*). Given that the site of covalent modification is located at the opening of the palmitate binding pocket, we analyzed the TEAD palmitate pockets from multiple crystal structures available in the Protein Data Bank (PDB) *Noland et al., 2016*; *Pobbati et al., 2015*. All the structures aligned well, indicating a conserved and relatively inflexible pocket. As shown in the *Figure 1b*, docking of MYF-01–37 and TEAD2 revealed that MYF-01–37 binds in a tunnel formed by side chains of hydrophobic residues F233, F251, V252, V329, V347, M379, L383, L387, F406, and F428. We also identified a side pocket lined with hydrophilic side chains contributed by S331, S345, S377, and Q410. We hypothesized that we could improve potency and selectivity for our covalent ligands by introducing polar interactions and shape complementarity to this side pocket. Therefore, we predicted that the optimized binding pocket is Y-shaped with the targeted cysteine at one end, and hydrophobic and hydrophilic pockets at the other two ends. We envisioned that occupying the hydrophilic side pocket will also provide chance to incorporate polar moiety, leading to reduced global hydrophobicity and more druglike molecules. With this rationale, a variety of hydrophilic groups were introduced to the pyrrolidine ring at *trans*-position relative to the trifluoromethyl phenyl moiety, leading to synthesis of a focused library of Y-shaped molecules, with representatives shown in *Figure 1c*. We employed an in vitro palmitoylation assay that uses an alkyne-tagged palmitoyl CoA as the lipidation substrate and clickable biotin (*Figure 1d*), to screen this series of compounds for their ability to inhibit palmitoylation (*Figure 1e*, *Figure 1—figure*

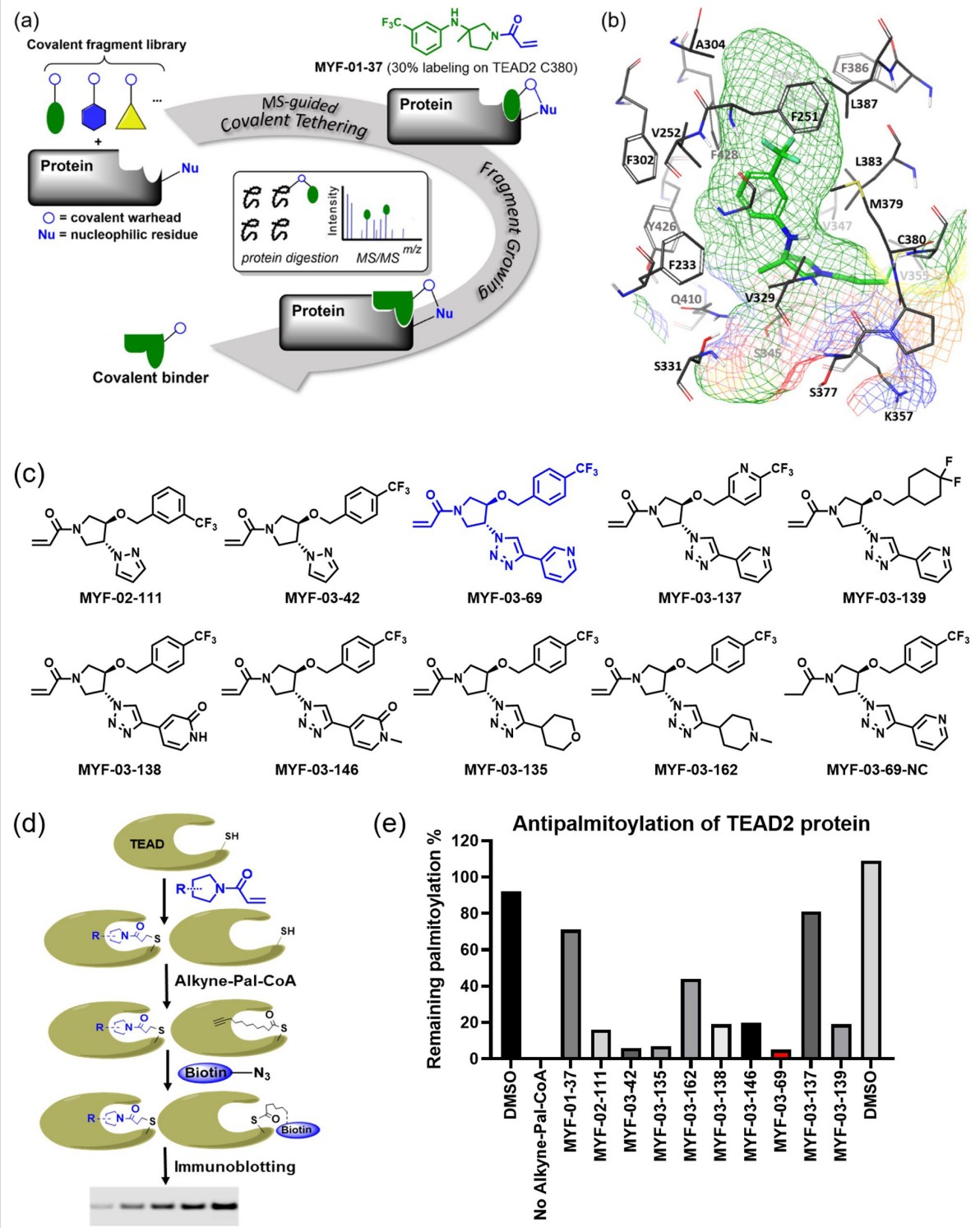

**Figure 1.** Covalent fragments screening and structure-guided design to identify Y-shaped compound MYF-03–69. (**a**) Illustration of covalent fragment library screening and optimization for TEAD inhibitor. (**b**) Surface of TEAD2 palmitate pocket depicted in mesh with MYF-01–37 (in green) modeled in. Residues forming the pocket were labeled. The color of mesh indicates hydrophobicity of the pocket surface. (**c**) Chemical structures of representative Y-shaped compounds. (**d**) The schematic diagram of in vitro palmitoylation assay. (**e**) Anti-palmitoylation activity of MYF-01–37 and Y-shaped compounds after TEAD2 protein was pre-incubated with 2 μM compound at 37 °C for 2 hr.

The online version of this article includes the following source data and figure supplement(s) for figure 1:

*Figure 1 continued on next page*

*Figure 1 continued*

**Source data 1.** Uncropped gel/blots and raw files related to *Figure 1*.

**Figure supplement 1.** Chemical structures of known TEAD palmitate pocket binders.

**Figure supplement 2.** Rationale of covalent fragments design.

**Figure supplement 3.** MYF-03–69 is the most potent TEAD inhibitor among derivatives that occupy a Y-shaped pocket.

*supplement 3*). We observed that the *para* substituted trifluoromethyl benzyl is superior to *meta* substituted trifluoromethyl benzyl (MYF-02–111 vs. MYF-03–42). We then replaced pyrazole by a substituted triazole to explore the hydrophilic pocket (MYF-03–135, MYF-03–162, MYF-03–138, MYF-03–146, MYF-03–69), and observed a sensitivity to various substitutions at C4 position of the triazole ring. For example, tetrahydropyran and pyridine enhanced the activities, whereas piperidine and pyridone were much less favored. In addition, we noticed that extending the hydrophobic tail of MYF-01–37 allowed more complete occupancy of the hydrophobic channel. Further modifications of the groups occupying the hydrophobic tunnel did not yield further improvement on the potency compared to trifluoromethyl benzyl (MYF-03–137, MYF-03–139), implicating restrict requirement for a hydrophobic moiety. Amongst this series of compounds, we identified MYF-03–69 (*Figure 1c*) as the most potent compound. The preferred stereochemistry on pyrrolidine ring was predicted by pocket analysis and docking results (*Figure 1—figure supplement 3*). To validate this, we compared the inhibitory activity of MYF-03–69 with its enantiomer MYF-03-69E in TEAD2 palmitoylation assay. The results indicated that compared with MYF-03–69, the enantiomer significantly lost activity at 2.5 µM concentration (*Figure 1—figure supplement 3*). Thus, we chose MYF-03–69 for further characterization.

## MYF-03-69 occupies the Y-shaped pocket and forms covalent bond with the conserved cysteine of TEAD

Given that labeling efficiency of the parental fragment MYF-01–37 was moderate, we examined whether the elaborated, Y-shaped molecule labeled TEAD more efficiently. Recombinant TEAD2 YBD was incubated with MYF-03–69, followed by mass spectrometry. Incubation with 10-fold molar excess of MYF-03–69 at room temperature for 1 hr led to 100% labeling (*Figure 2a*) and follow-up trypsin digestion and MS/MS analysis confirmed C380 as the modified residue (*Figure 2b*). This covalent bond formation and the exact binding mode was further validated by obtaining a high-resolution co-crystal structure of TEAD1 YBD in complex with MYF-03–69 (1.68 Å) (*Figure 2c*, *Supplementary file 1*). Unambiguous electron density of a covalent bond was observed between TEAD1 C359 (analogous to C380 in TEAD2) and the acrylamide warhead of MYF-03–69 with the remaining part of the molecule adopting a Y-shape and establishing specific interactions with the hydrophobic pocket and hydrophilic patch as predicted (*Figure 2c*). MYF-03–69 was completely buried in the lipid pocket with the *para*-trifluoromethyl benzyl group extended along a lipid trajectory, forming extensive hydrophobic interactions with side chains of A223, F239, V240, A292, I366, and F407. The 3-pyridinyl group occupied the side pocket, forming a favorable water-bridged hydrogen bond network with adjacent S310 and Y312 via pyridinyl nitrogen (*Figure 2d*). Overall, high-resolution structural analysis confirms that MYF-03–69 forms covalent attachment with the TEAD cysteine that is the site of *S*-palmitoylation, and that the elaborated molecule forms specific interactions via both hydrophobic and hydrophilic regions surrounding the targeted cysteine which differs from the reported covalent TEAD inhibitor K-975 (*Figure 1—figure supplement 3*).

To further characterize biochemical activity of MYF-03–69, we performed a dose titration for all TEAD paralogs (*Figure 2e*, *Figure 2—figure supplement 1*). Preincubation of all recombinant TEAD1-4 YBDs with MYF-03–69 potently inhibited palmitoylation with similar $IC_{50}$ values at submicromolar concentrations, indicating that MYF-03–69 is a pan-TEAD inhibitor. This is consistent with the high sequence conservation of residues in the palmitate pockets of TEAD1-4. In contrast, MYF-03–69-NC (*Figure 1c*), the non-covalent counterpart of MYF-03–69 incapable of forming covalent bond, completely lost activity across all TEADs, which demonstrated the essentiality of covalent bond formation.

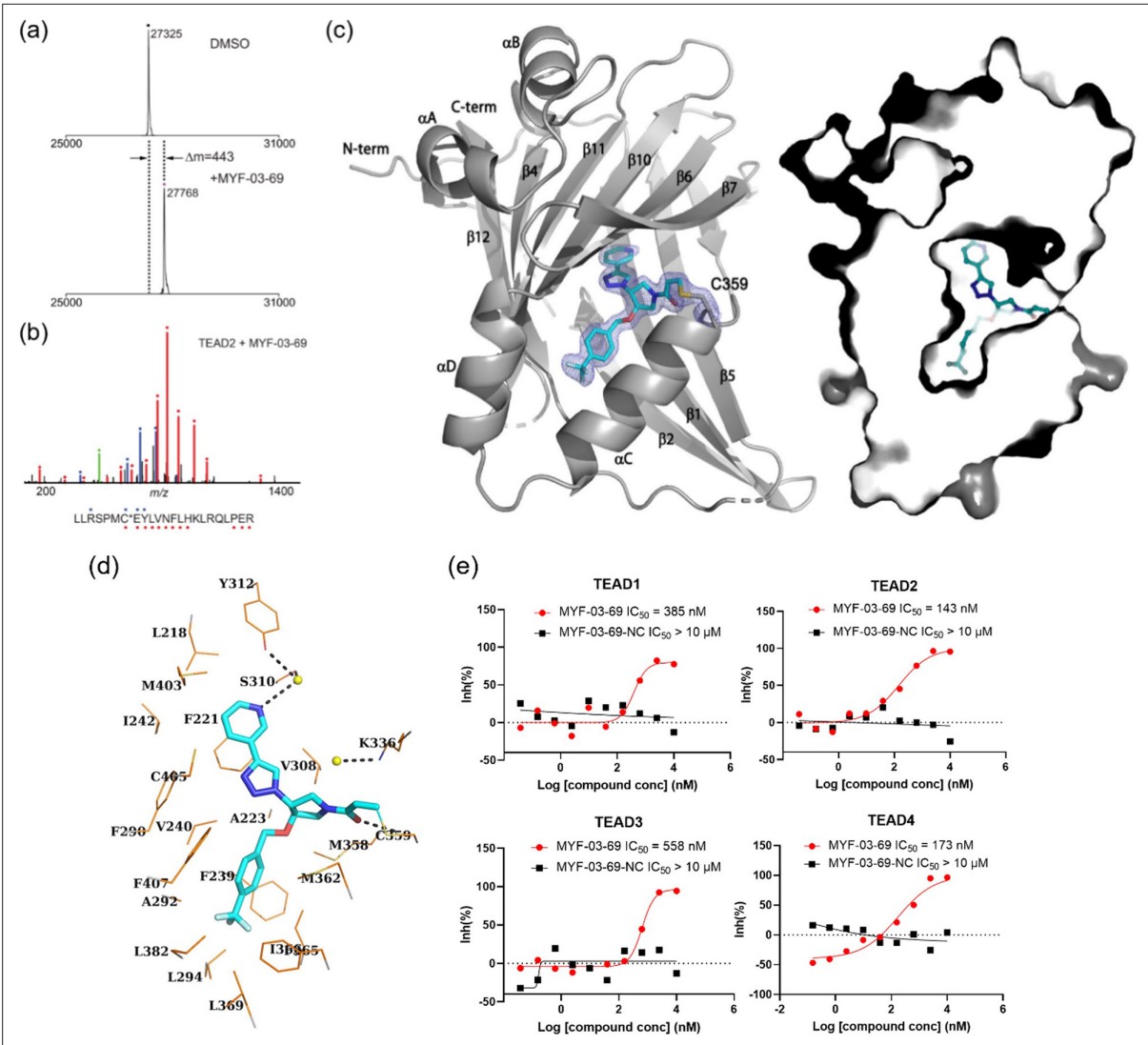

**Figure 2.** MYF-03–69 binds in TEAD palmitate pocket covalently through the conserved cysteine. (**a**) The mass labeling of intact TEAD2 protein by MYF-03–69. (**b**) Trypsin digestion and tandem mass spectrum (MS/MS) localize labeling site of cysteine 380. (**c**) Co-crystal structure of MYF-03–69 with TEAD1 indicates covalent bond formation with Cys359. The compound adopts Y-shape and binds in both lipid tunnel and hydrophilic side pocket. (**d**) Interactions between MYF-03–69 and TEAD1 palmitate pocket. (**e**) A dose titration of MYF-03–69 and MYF-03–69-NC in anti-palmitoylation assay on TEAD1-4. Recombinant YBD protein of TEADs were preincubated with compounds at 37 °C for 2 hr. Data are representative of n=3 independent experiments.

The online version of this article includes the following source data and figure supplement(s) for figure 2:

**Source data 1.** Uncropped gel/blots and raw files related to *Figure 2*.

**Figure supplement 1.** Original western blot images of palmitoylation assays in YBD protein of TEAD1-4.

## MYF-03-69 inhibits TEAD palmitoylation and disrupts endogenous YAP-TEAD association in cells

We next investigated the ability of MYF-03–69 to modulate TEAD palmitoylation inside cells with $\omega$-alkynyl palmitic acid (Alkyne-C16) as probe. Alkyne-C16 contains an alkyne group at C15 and has been shown to be metabolically incorporated into cellular proteins at native palmitoylation sites. The previously reported pulse-chase style experiment *Holden et al., 2020* was adopted and modified to monitor TEAD palmitoylation in HEK293T cells transfected with Myc-TEAD4. After the cells were incubated with Alkyne-C16 for 24 hr, the labeled Myc-TEAD4 was conjugated with biotin-azide through a Cu(I)-catalyzed click reaction. With western-blotting, we found that the alkyne-palmitate

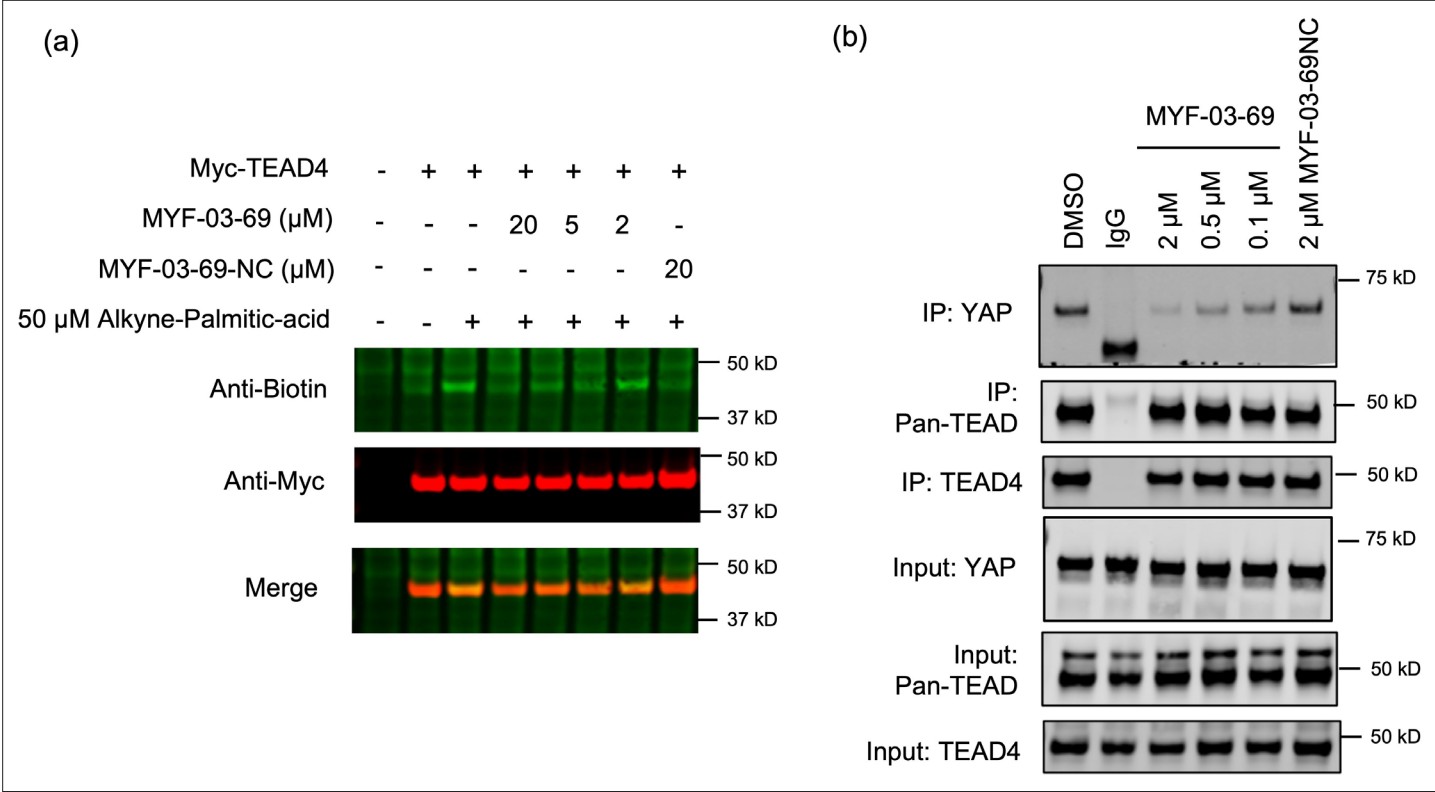

**Figure 3.** MYF-03–69 inhibits palmitoylation of TEAD protein and disrupts its association with YAP in cells. (**a**) Palmitoylation of Myc-TEAD4 in HEK293T cells after treatment with MYF-03–69 and MYF-03–69-NC indicated by an alkyne-palmitic-acid probe and click chemistry. Cells were treated for 24 hr. (**b**) Co-immunoprecipitation (Co-IP) of endogenous YAP and TEAD in NCI-H226 cells after treatment with MYF-03–69 and MYF-03–69-NC at indicated doses. Cells were treated for 24 hr.

The online version of this article includes the following source data and figure supplement(s) for figure 3:

**Source data 1.** Uncropped gel/blots and raw files related to *Figure 3*.

**Figure supplement 1.** MYF-03–69 exhibited low reactivity towards the cysteine proteome.

labeling of Myc-TEAD4 decreased after cotreatment with varying concentrations of MYF-03–69 for 24 hr compared to the DMSO treatment, while TEAD protein levels were not affected (*Figure 3a*). This result suggested that MYF-03–69 competed off palmitic acid during TEAD lipidation in cells. Consistently with biochemical result, the negative control compound MYF-03–69-NC did not exhibit any effect on TEAD (*Figure 3a*).

Palmitoylation of TEADs was recently proposed to be required for their binding to YAP and TAZ[13b]. Although TEAD inhibitors with different chemotypes have been recently developed, it remains controversial whether pharmacological targeting of the TEAD palmitate pocket could disrupt YAP/TAZ association *Pobbati and Rubin, 2020*. In order to investigate whether MYF-03–69 impairs YAP-TEAD interaction, we conducted an endogenous co-immunoprecipitation (co-IP) experiment to monitor YAP-TEAD interactions in the presence of the compound. Notably, 24 hr treatment of NCI-H226 cells with MYF-03–69 significantly decreased endogenous YAP co-immunoprecipitation with a pan-TEAD antibody, whereas this treatment had minimal effect on TEAD protein level (*Figure 3b*). In parallel, we also evaluated MYF-03–69-NC and observed no effect on YAP-TEAD association in cells. Taken together, these results suggest that MYF-03–69-mediated perturbation of TEAD palmitoylation disrupts YAP and TEAD protein interactions.

To demonstrate MYF-03–69 as a selective covalent TEAD binder, we interrogated the proteome-wide reactivity profile of MYF-03–69 on cysteines using a well-established streamlined cysteine activity-based protein profiling (SLC-ABPP) approach (*Figure 3—figure supplement 1*; *Kuljanin et al., 2021*). We employed the cysteine reactive desthiobiotin iodoacetamide (DBIA) probe *Kuljanin et al., 2021* which was reported to map more than 8000 cysteines and performed a competition study

in NCI-H226 cells pretreated with 0.5, 2, 10, or 25 µM of MYF-03–69 for 3 hr in triplicate. The cysteines that were conjugated >50% (competition ratio CR >2) compared to DMSO control were analyzed and assigned to the protein targets (*Figure 3—figure supplement 1*). In the DMSO control group, although DBIA mapped 12,498 cysteines in total, the TEAD PBP cysteines were not detected. Among 12,498 mapped cysteines, only 7 cysteines were significantly labeled (i.e. exhibited >50% conjugation or CR >2) by 25 µM of MYF-03–69, and all these sites exhibited dose-dependent engagement (*Supplementary file 2*). The missing TEAD cysteines might be due to low TEAD1-4 protein abundance and/or inability of the PBP cysteines to be labeled by DBIA. DBIA probe can only be applied in cell lysate context, which might also result in the missing labeling on TEAD. To our knowledge, all the proteins with meaningful labeling efficiency (CR >2) cysteine sites are not known to be involved in the YAP/TAZ-TEAD signaling pathway. Taken together, our proteomic analysis suggests that MYF-03–69 exhibited quite low reactivity towards the cysteine proteome, although TEAD PBP cysteines were not detected by the DBIA probe.

## MYF-03-69 inhibits TEAD transcription and downregulates YAP target genes expression in mesothelioma cells

The majority of mesothelioma patients (50–75%) harbor genetic alterations in Hippo pathway regulatory components, including *NF2* loss of function mutation/deletion and *LATS1-PSEN1* fusion/*LATS2* deletion, which lead to YAP activation and Hippo pathway gene expression *Bueno et al., 2016*. Thus, in order to monitor YAP-TEAD transcriptional activity, Hippo/YAP reporter cells were generated in NCI-H226, a *NF2*-deficient MPM cell line. After 72 hr treatment of MYF-03–69, YAP-TEAD transcriptional activity of reporter cells significantly decreased in a dose-dependent manner with an $IC_{50}$ value of 56 nM, while MYF-03–69-NC had minimal effect (*Figure 4a*). The transcriptional inhibition also led to significant downregulation of canonical YAP target genes including *CTGF*, *CYR61*, *IGFBP3*, *KRT34*, and *NPPB* and upregulation of proapoptotic gene *BMF* in NCI-H226 cells and MSTO-211H, a *LATS1-PSEN1* fusion MPM cell line (*Figure 4b–c*), while it showed much milder effects on transcription in normal mesothelium cells MeT-5A (*Figure 4—figure supplement 1*). Protein level of *CYR61* and *AXL* gene products also decreased (*Figure 4d*, *Figure 4—figure supplement 2*). Encouraged by these results, we set out to study whole transcriptome perturbation by MYF-03–69. RNA sequencing was performed in NCI-H226 cells that were treated with 0.1 µM, 0.5 µM, and 2 µM of MYF-03–69. There were 339 genes that exhibited significant differential expression at 2 µM treatment, and the majority of them changed in a dose-dependent manner (*Figure 4e* and *Supplementary file 3*). The genes that were differentially expressed with statistical significance (Fold change ≥ 1.5 and adjusted p value ≤ 0.05) at 2 µM treatment condition, are colored in red and partially labeled in the volcano plot (*Figure 4f*). Compared to high concentration, the number of genes with significantly altered expression level dropped off quickly at lower concentrations (*Figure 4—figure supplement 3*). For example, at the concentration of 0.1 µM, only *CTGF* showed statistically significant change (*Figure 4—figure supplement 3*). While multiple YAP target genes such as *CTGF*, *ADM*, *ANKRD1* were significantly downregulated *Zhao et al., 2008*, *DDIT4* was observed to be upregulated *Kim et al., 2015*. To investigate whether the 339 differentially expressed genes were concentrated in particular biological pathways, KEGG pathway enrichment calculation *Zhou et al., 2019* were carried out, which demonstrated Hippo signaling pathway was among the top 5 enriched processes (*Figure 4g*). Beyond the Hippo pathway, gap junction *Karaman and Halder, 2018* and WNT signaling pathway *Li et al., 2019* were also enriched, consistent with pleiotropic functions and crosstalk of YAP-TEAD pathway in diverse biological processes. Overall, we documented that MYF-03–69 affects transcription in a manner consistent with its activity as a disruptor of YAP-TEAD interactions.

## MYF-03-69 selectively inhibits mesothelioma cancer cells with defective Hippo signaling

Next, we investigated anti-cancer activity of MYF-03–69 on Hippo signaling defective mesothelioma cells. As shown in *Figure 5a* and *Figure 5—figure supplement 1*, 5-day cell growth assays demonstrated that MYF-03–69 potently retarded the cell growth of NCI-H226 and MSTO-211H, while it showed no antiproliferation activity against MeT-5A and NCI-H2452 cells, which are non-cancerous mesothelium cells and mesothelioma cells with intact Hippo signaling, respectively. These antiproliferative effects in NCI-H226 and MSTO-211H cells were observed under 3D spheroid suspensions

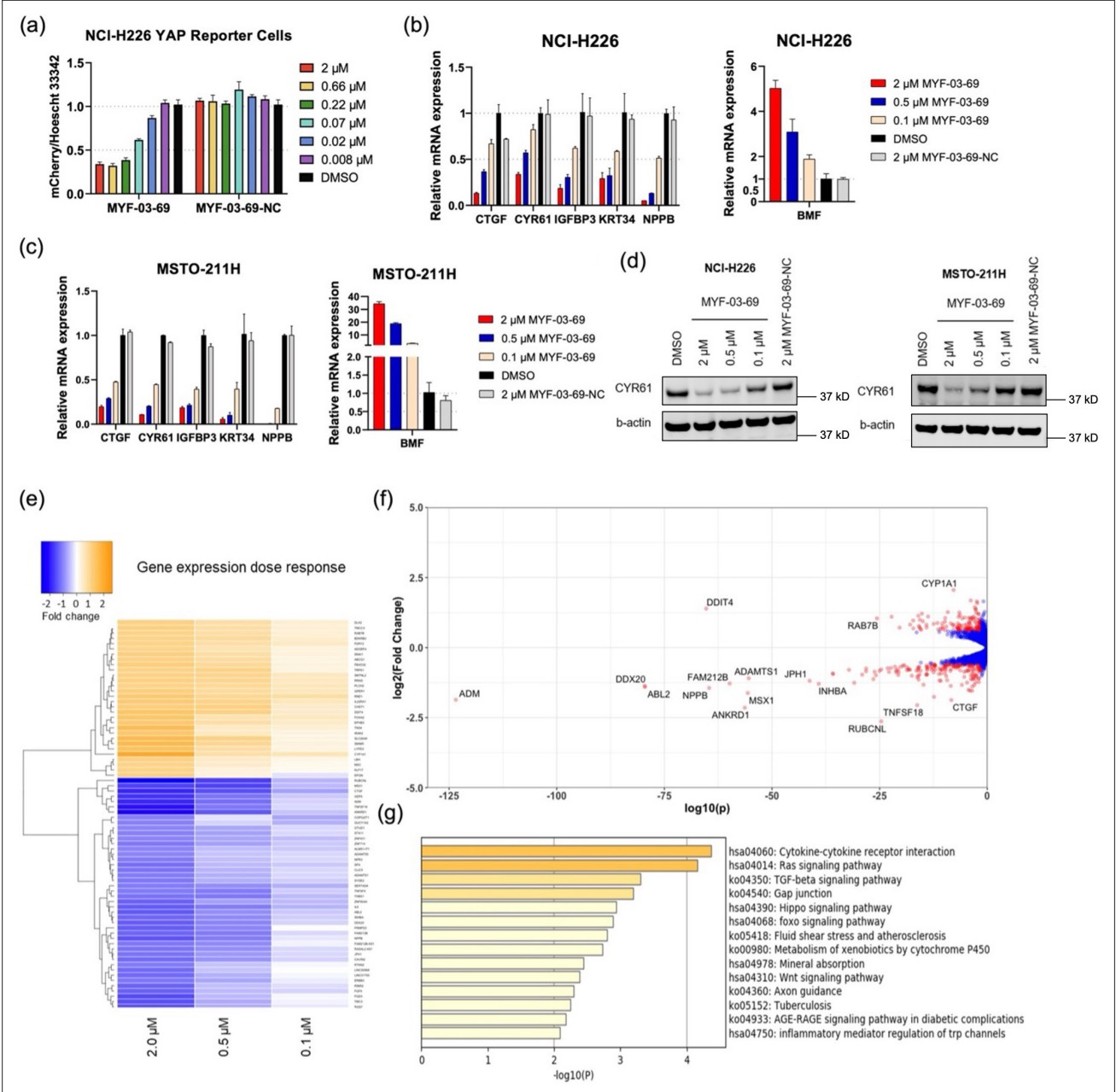

**Figure 4.** MYF-03–69 inhibits YAP-TEAD transcription. (**a**) MYF-03–69, but not MYF-03–69-NC, inhibits YAP-TEAD transcriptional activity in NCI-H226 mCherry reporter cells. Cells were treated for 72 hr. Data were presented as mean ± SD of n=3 biological independent samples. (**b–c**) MYF-03–69, but not MYF-03–69-NC, downregulates YAP target genes and upregulates a pro-apoptotic gene *BMF*. Cells were treated for 24 hr. Data were presented as mean ± SD of n=3 biological independent samples. (**d**) MYF-03–69, but not MYF-03–69-NC, downregulates CYR61 protein level in NCI-H226 and MSTO-211H cells. (**e**) Heatmap for gene expression change with MYF-03–69 treatment at indicated concentrations. (**f**) Differential gene expression from RNA sequencing of NCI-H226 cells treated with 2 µM MYF-03–69. The differentially expressed genes with FC ≥1.5 and p ≤ 0.05 were colored in red and labeled. Not all differentially expressed genes were labeled. (**g**) Pathway enrichment analysis of differentially expressed genes from 2 µM compound treatment samples.

The online version of this article includes the following source data and figure supplement(s) for figure 4:

**Source data 1.** Uncropped gel/blots and raw files related to *Figure 4*.

**Figure supplement 1.** MYF-03–69 showed comparatively weak inhibition on YAP-TEAD downstream gene expression in normal mesothelium cell MeT-5A upon 6 hr and 24 hr treatment.

**Figure supplement 2.** MYF-03–69 downregulated product of canonical YAP downstream gene AXL with minimal effect on TEAD stability.

**Figure supplement 3.** The number of genes with significantly altered expression level decreased with lower concentration treatment in NCI-H226 cells.

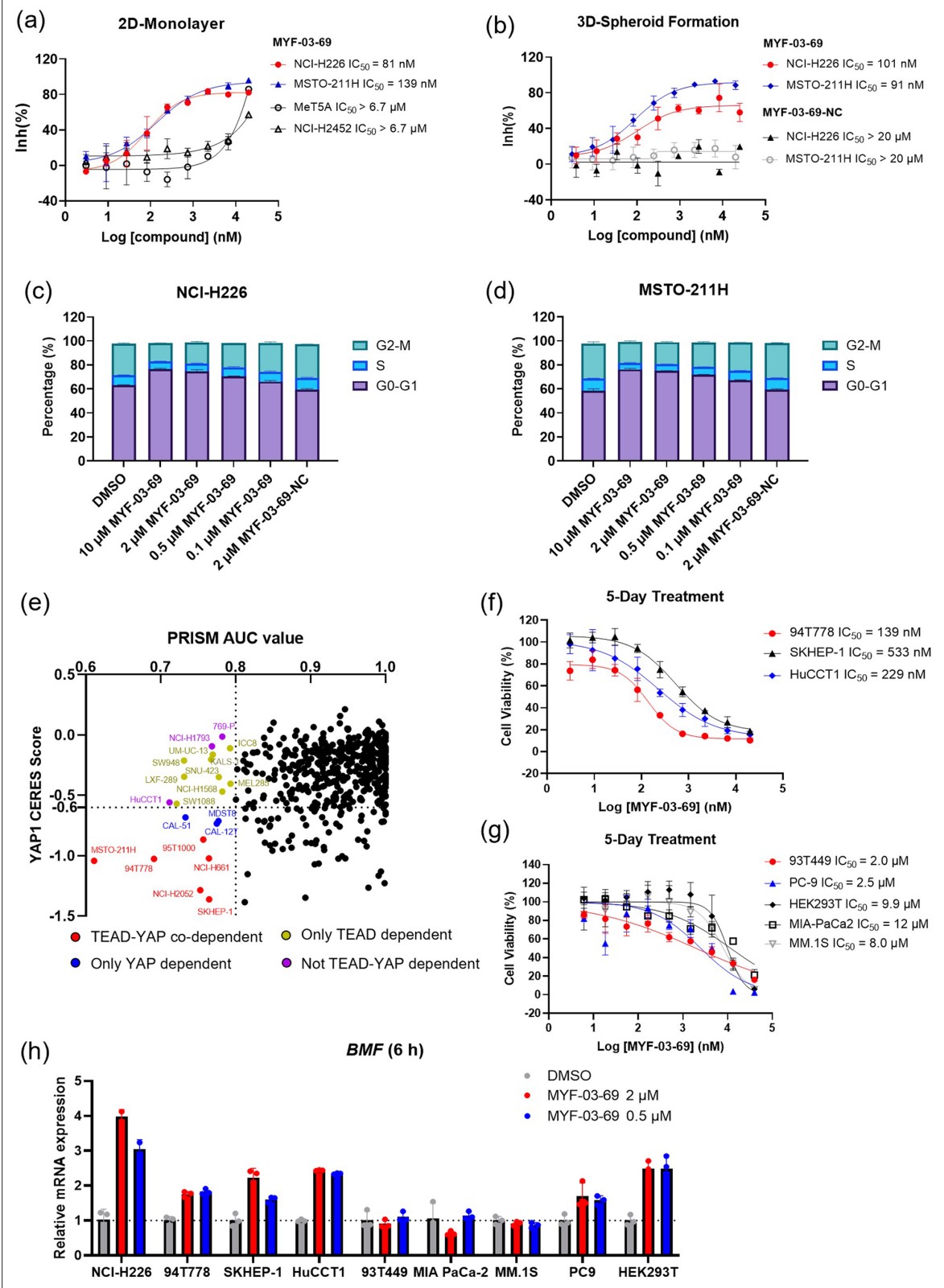

**Figure 5.** MYF-03–69 selectively inhibits proliferation of cancer cells with YAP or TEAD dependency regardless of lineage. (**a**) Antiproliferation $IC_{50}$ curve of MYF-03–69 on three mesothelioma cell lines (NCI-H226, MSTO-211H and NCI-H2452) and a normal noncancerous mesothelium cell line (Met-5A). Data were presented as mean ± SD of n=3 biological independent samples. (**b**) Antiproliferation $IC_{50}$ curve of MYF-03–69 and MYF-03–69-NC in 3D cell culture. Data were presented as mean ± SD of n=3 biological independent samples. (**c–d**) Cell cycle arrest induced by MYF-03–69, but not MYF-03–69-

*Figure 5 continued on next page*

*Figure 5 continued*

NC. Cells were treated for 48 hr at indicated doses. Data were presented as mean ± SD of n=3 biological independent samples. (**e**) PRISM profiling across a broad panel of cell lineages. 903 cancer cells were treated with MYF-03–69 for 5 days. The viability values were measured at 8-point dose manner (3-fold dilution from 10 μM) and fitted a dose-response curve for each cell line. Area under the curve (AUC) was calculated as a measurement of compound effect on cell viability. CERES score of YAP1 or TEADs from CRISPR (Avana) Public 21Q1 dataset (DepMap) were used to estimate gene-dependency. Cell lines without CERES Score of YAP1 were excluded from the figure. The CERES Score of most dependent TEAD isoform was used to represent TEAD dependency. Cell lines with a dependency score less than –0.6 were defined as the dependent cell lines. (**f**) Antiproliferation curves of cell lines that are sensitive to MYF-03–69 treatment besides mesothelioma. Data were presented as mean ± SD of n=3 biological independent samples. (**g**) Antiproliferation curves of cell lines that are insensitive to MYF-03–69 treatment. Data were presented as mean ± SD of n=3 biological independent samples. (**h**) *BMF* expression level after treatment with 0.5 or 2 μM MYF-03–69 over 6 hr in different cells. Data were presented as mean ± SD of n=3 biological independent samples.

The online version of this article includes the following figure supplement(s) for figure 5:

**Figure supplement 1.** MYF-03–69 inhibited the proliferation of TEAD-dependent cancer cells.

culture condition as well (*Figure 5b*). Under the same conditions, the negative control compound MYF-03–69-NC was not antiproliferative (*Figure 5b*, *Figure 5—figure supplement 1*). Further, cell cycle analysis demonstrated that 48 hr treatment with MYF-03–69 on NCI-H226 and MSTO-211H cells caused cell cycle arrest at the G1 phase, which is in accordance with previous findings obtained from genetic knockdown of YAP *Mizuno et al., 2012*, while negative control compound had no effect (*Figure 5c–d*). Collectively, these results show that inhibition of TEAD palmitoylation by MYF-03–69 effectively and precisely affects YAP-TEAD function.

## MYF-03-69 inhibits YAP or TEAD-dependent cancer cells beyond mesothelioma

To further investigate whether inhibition by MYF-03–69 was selectively lethal to YAP/TEAD-dependent cancers, 903 barcoded cancer cell lines were screened using the PRISM assay *Yu et al., 2016*. As shown in *Figure 5e and a* small portion of cell lineages exhibited vulnerability (*Supplementary file 4*). Correlation analysis reveals that the dependency scores of *TEAD1* and *YAP1* according to genomic knockout dataset (DepMap) provided the highest correlation with the sensitivity profile (*Figure 5— figure supplement 1*, *Supplementary file 5*). This is followed by *TP53BP2*, a gene that is also involved in Hippo pathway as activator of TAZ *Liu et al., 2011*. For example, when we used a threshold of AUC ≤0.8 for the sensitivity to MYF-03–69, 33 cell lines were selected and the majority of which are either YAP- or TEADs-dependent cells, as suggested by CERES scores (*Figure 5e*). These include YAP/TEAD co-dependent cells (red dots), YAP-dependent cells (blue dots), and TEAD-dependent cells (yellow dots). Next, to verify the antiproliferative activity, we chose three sensitive cell lines (94T778, SKHEP-1 and HuCCT1) and three insensitive cell lines (93T449, MIA PaCa-2 and MM.1S) indicated by PRISM screening, as well as two additional cell lines (PC9, HEK293T) that were not included in PRISM screening panel to test in 5-day antiproliferation assay. As shown in *Figure 5f*, liposarcoma cell 94T778, hepatic adenocarcinoma cell SKHEP-1 and cholangiocarcinoma cell HuCCT1 were inhibited with nanomolar IC$_{50}$. In contrast, liposarcoma cell 93T449, pancreatic ductal adenocarcinoma cell MIA PaCa-2 and myeloma cell MM.1S were barely inhibited (*Figure 5g*). Since proapoptotic gene *BMF* was known to be released from repression upon TEAD inhibition *Kurppa et al., 2020*, we examined *BMF* mRNA levels after 6 hr treatment with 0.5 and 2 μM MYF-03–69 on both sensitive and insensitive cells. As expected, *BMF* levels increased in three sensitive cells, but remained unchanged in three insensitive cells (*Figure 5h*). We also noted the upregulation of *BMF* in PC9 and HEK293T cells which are resistant to TEAD inhibition (*Figure 5g–h*). A heterogeneity of signaling pathway dependency, especially EGFR signaling and YAP1 activation in PC9 cells might account for the insensitivity to TEAD inhibition alone. Taken together, we demonstrate that TEAD inhibition could be an exploitable vulnerability across multiple malignant tumor models besides mesothelioma and that upregulation of *BMF* gene is a common phenomenon in those cancer cells with sensitive antiproliferative response to TEAD inhibitor.

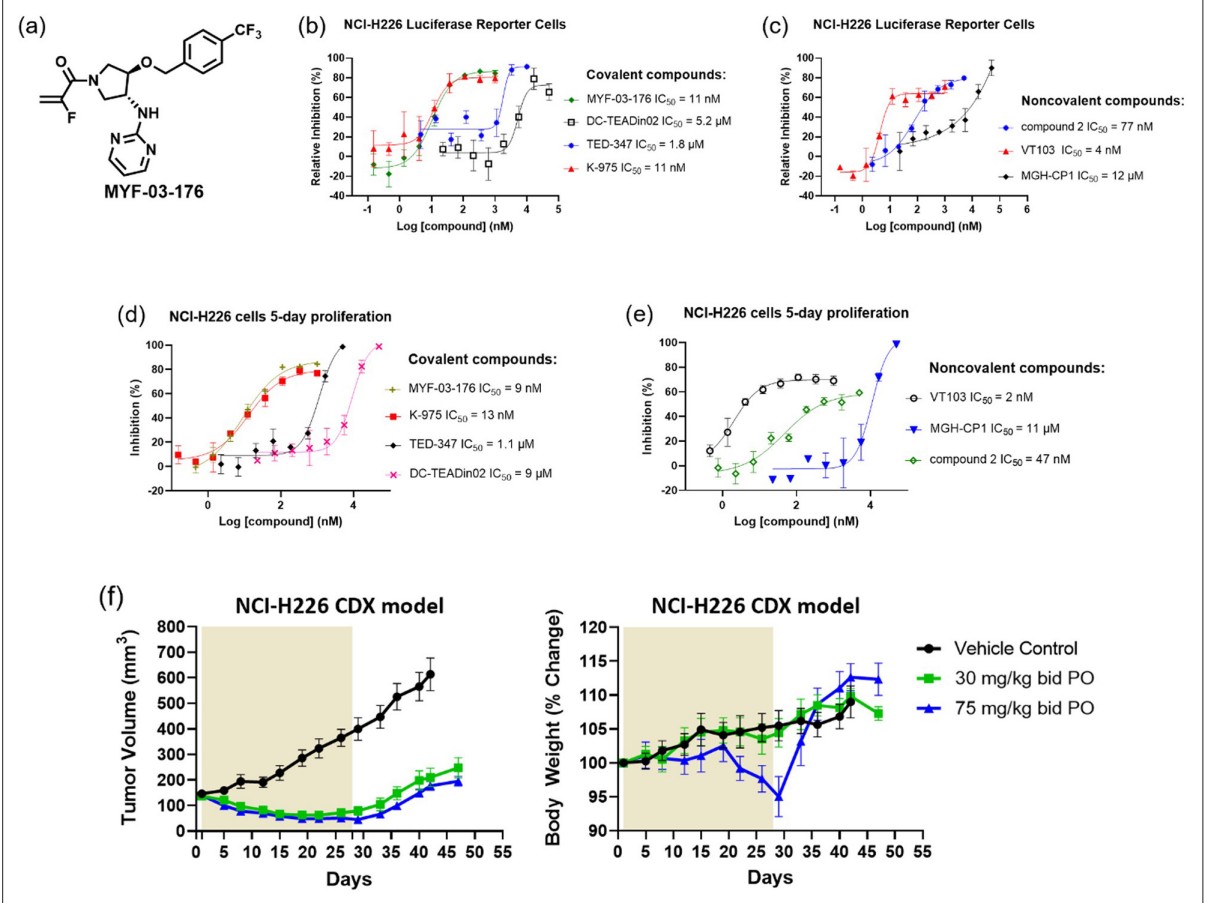

**Figure 6.** MYF-03–176 is a potent and orally bioavailable YAP-TEAD transcription inhibitor and suppresses tumor growth in mesothelioma xenograft mouse model. (**a**) Chemical structure of MYF-03–176. (**b–c**) Inhibitory effect of MYF-03–176 and other TEAD PBP binders on YAP-TEAD transcription in NCI-H226 luciferase reporter cells. Cells were treated for 72 hr. Data were presented as mean ± SD of n=3 biological independent samples. (**d-e**) Antiproliferation effect of MYF-03–176 and other TEAD PBP binders in NCI-H226 cells. Cells were treated for 5 days. Data were presented as mean ± SD of n=3 biological independent samples. (**f**) In vivo efficacy of MYF-03–176 in NCI-H226 CDX mouse model (n=8–9 per group).

The online version of this article includes the following figure supplement(s) for figure 6:

**Figure supplement 1.** $^1$H NMR, $^{13}$C NMR spectra for MYF-03–69.

**Figure supplement 2.** $^1$H NMR, $^{13}$C NMR spectra for MYF-03–176.

**Figure supplement 3.** MYF-03–69 and MYF-03–176 inhibited TEAD transcriptional activity.

### MYF-03-176 exhibits significant antitumor efficacy in NCI-H226 xenograft mouse model through oral administration

In order to demonstrate therapeutic potential of covalent TEAD palmitoylation disruptor, extensive medicinal chemistry efforts were undertaken on this Y-shaped scaffold, leading to a more potent and more importantly, orally bioavailable MYF-03–69 analog MYF-03–176 (*Figure 6a*, *Figure 6—figure supplements 1–2*). Before we administrated the mice with MYF-03–176, we conducted a head-to-head comparison on the activity with MYF-03–69 as well as other reported TEAD inhibitors including K-975, compound 2 and VT103. We used a stably transfected TEAD luciferase reporter system in NCI-H226 cells to profile transcriptional effect of TEAD inhibitors. Consistently with the results of mCherry reporter system, MYF-03–69 inhibited the transcription of reporter gene at $IC_{50}$ of 45 nM (*Figure 6—figure supplement 3*) and interestingly MYF-03–176 was 3-fold more potent with $IC_{50}$ of 11 nM (*Figure 6b*). Further, similar to MYF-03–69, MYF-03–176 led to a significant downregulation of YAP target genes *CTGF*, *CYR61*, *ANKRD1* and an upregulation of *BMF* (*Figure 6—figure supplement 3*). As comparison, MYF-03–176 exhibited comparable and even better activity than K-975 and compound 2, while was less potent than VT103 as shown in *Figure 6b~c*. The antiproliferative

effect of MYF-03–176 in NCI-H226 cells was similar to VT-103 and K-975 but even stronger than MYF-03–69 and compound 2. Although K-975 displayed similar activities in the anti-proliferation assay, it has extremely short half-life compared with MYF-03–176 (*Supplementary file 6*). With the support of all these data from MYF-03–176 and decent pharmacokinetics properties including low clearance and high oral bioavailability (*Supplementary file 7*), we then administrated MYF-03–176 to NCI-H226 cell line derived xenograft (CDX) mouse model. The tumor-bearing mice were randomized and orally administrated MYF-03–176 twice daily for 28 days. Significant antitumor activity with tumor regressions was observed at both 30 mg/kg (average tumor regression of 54%) and 75 mg/kg (average tumor regression of 68%). The anti-tumor activity at the two doses was comparable (p=0.23, 2-way ANOVA). The 30 mg/kg dose was well tolerated with an average body weight gain comparable to vehicle control (data not shown), while at the 75 mg/kg dose, an average wight loss of 5% was observed. However, at this 75 mg/kg dose, three of eight animals demonstrated 12–14% body weight loss. The weight loss recovered once drug administration was stopped (*Figure 6f*). Taking all these together, MYF-03–176 potently inhibited Hippo signaling defective MPM cells and exhibited strong antitumor effect in the human mesothelioma NCI-H226 CDX model in vivo, representing a promising leading compound for drug discovery.

## Discussion

As widely recognized oncogenic proteins, YAP/TAZ have emerged as potentially attractive targets for anti-cancer drug development. However, the unstructured nature of YAP/TAZ proteins renders them difficult to target by conventional occupancy-based small molecules. Therefore, the majority of compounds that are known to inhibit YAP activity are targeted at upstream stimulators of YAP/TAZ. In this context, TEADs, key components of Hippo signaling pathway that depend on YAP/TAZ binding for activation of transcriptional activity, have attracted attention. Here, the presence of a well-defined palmitate binding pocket (PBP) on TEADs suggested opportunity for small molecule inhibitor development. However, recent studies have not fully resolved the question of whether occupying TEAD PBP disrupts or stabilizes YAP-TEAD interaction. Chan et al. suggested that palmitoylation of TEAD stabilizes YAP-TEAD interaction[13b]. Holden et al. reported the reversible PBP binder compound 2 had minimal to no disruptive effect on YAP-TEAD interaction biochemically but transformed TEAD into a dominant-negative transcriptional repressor *Holden et al., 2020*. VT series compounds reported by Tang et al. dissociated endogenous YAP-TEAD according to co-IP experiment *Tang et al., 2021*, which is consistent with our result of MYF-03–69. Other data from MGH-CP1 *Li et al., 2020* and covalent binder K-975 *Kaneda et al., 2020* pointed to the same conclusion while the experiments were performed with exogenous YAP or TEAD. K-975 and TED-347 were reported to disrupt protein-protein interaction between TEAD and YAP peptide in biochemical assay. However, their co-crystal structures indicated no conformation change compared with palmitoylated TEAD. Similarly, superimposition of co-crystal structures of TEADs with MGH-CP1, compound 2, VT105, MYF-03–69 and their corresponding palmitoylated TEADs did not reveal any obvious conformation change or side chain move that might affect YAP binding. Taken all these together, the transcriptional inhibitory effect of these TEAD PBP binders might result from disruption of certain dynamic process of TEAD lifecycle. Recently, several reports elaborated that YAP/TAZ and TEAD undergo lipid-lipid phase separation (LLPS) during transcription process *Cai et al., 2019*. One hypothesis is the alkyl chain of palmitoyl might escape from TEAD PBP and expose to outside when forming such functional transcription compartment. The hydrophobic nature of the lipid chain may play a role in organizing the disordered and hydrophobic region of YAP. Thus, replacement of palmitate by these PBP binders leads to incapability of TEAD to organize into transcription machinery, although the direct binding to YAP peptide may not being affected in vitro. The underlying mechanism is remained to be uncovered in the future.

MYF-03–69 is a covalent compound that we designed to target the conserved cysteine on TEAD, the site of palmitoylation. The starting point for MYF-03–69 was a fragment hit that we identified through screening of a biased covalent fragment library and further optimized using

structure-based strategy. Unlike screening reversible fragments which often require significant follow-up efforts to determine binding site, the MS-guided covalent tethering method gives unambiguous information of labeling site in a cost and time efficient way. Unlike previously reported compounds that engage only the hydrophobic PBP, our optimized inhibitor was developed to exploit interactions with not only the PBP but a hydrophilic binding pocket that we have identified during this study. Thus, our optimized inhibitor MYF-03–69 is a Y-shaped molecule which efficiently occupies two pockets, as well as covalently binds the conserved cysteine, which contributed significantly to both potency and specificity. Therefore, we speculated that MYF-03–69 may be employed as a useful tool to interrogate the process of how PBP binder affect TEAD homeostasis. We also propose that further optimization of the basic Y-shaped molecule reported here may yield paralog-selective TEAD inhibitors.

Aberrant regulation of Hippo-YAP/TAZ-TEAD signaling axis has been recognized as driver for genesis and development of multiple cancers, especially in mesothelioma. Besides the Hippo pathway, diverse signaling pathways such as Wnt, TGF-β, EGFR and Hedgehog pathways can also potentiate the activation of YAP/TAZ. Given such essentiality, it is important to ascertain what portion of these YAP/TAZ activated cancers are vulnerable to TEAD inhibition. The pan-TEAD selective nature of MYF-03–69 allowed us to examine this question at the TEAD family level using a panel of more than 900 cancel cell lines. Our results demonstrated that MYF-03–69 exhibited selective antiproliferative effect on YAP or TEAD dependent cancer cells from different lineages including mesothelioma, liver cancer, liposarcoma, and lung cancer. Strong antitumor efficacy in human mesothelioma NCI-H226 CDX model has been achieved with an orally bioavailable compound MYF-03–176 in the same scaffold, which is ready to be tested in various cancer models. Moreover, we noted that increased expression of a proapoptotic *BMF* gene correlates with response to TEAD inhibition in some cancer cell lines.

Collectively, this study provides evidence that MYF-03–69 represents a potent, covalent, cysteine-targeted pan-TEAD inhibitor that disrupts YAP-TEAD association and affects transcription in covalent binding-dependent manner. Given its 'mild' covalent warhead and low reactivity across proteome, we nominate MYF-03–69 as a viable lead for drug discovery for not only mesothelioma, but other YAP or TEAD-dependent cancers such as liver cancer, liposarcoma, and lung cancer.

## Materials and methods

**Key resources table**

| Reagent type (species) or resource | Designation | Source or reference | Identifiers | Additional information |
|---|---|---|---|---|
| Cell line (*Homo sapiens*) | NCI-H226 (Mesothelioma, Male) | ATCC | CRL-5826 | |
| Cell line (*Homo sapiens*) | MSTO-211H (Mesothelioma,Male) | ATCC | CRL-2081 | |
| Cell line (*Homo sapiens*) | NCI-H2452 (Mesothelioma, Male) | ATCC | CRL-5946 | |
| Cell line (*Homo sapiens*) | MeT-5A (Mesotheliumcells) | ATCC | CRL-9444 | |
| Antibody | Anti-human YAP (Rabbit monoclonal) | Cell Signaling | Cat#14074 S | WB (1:1000) |
| Antibody | Anti-human pan-TEAD (Rabbit monoclonal) | Cell Signaling | Cat#13295 S | WB (1:1000) |
| Antibody | Anti-human TEAD4 (Mouse monoclonal) | Abcam | Cat#ab58310 | WB (1:1000) |
| Antibody | Anti-human b-actin (Mouse monoclonal) | Cell Signaling | Cat#3700 S | WB (1:2000) |
| Recombinant DNA reagent | Myc-TEAD4 (plasmid) | Addgene | RRID: Addgene_24638 | Overexpression of Myc-TEAD4 |
| Recombinant DNA reagent | TBS-mCherry vector (plasmid) | Cancer Cell. 2020 Jan 13; 37(1): 104–122.e12 | | Monitor TEAD transcriptional activity |
| Commercial assay or kit | Dynabeads Co-Immunoprecipitation Kit | ThermoFisher Scientific | Cat#14321D | |

*Continued on next page*

*Continued*

| Reagent type (species) or resource | Designation | Source or reference | Identifiers | Additional information |
|---|---|---|---|---|
| Commercial assay or kit | SuperScript III First-Strand Synthesis System Kit | ThermoFisher Scientific | Cat#18080051 | |
| Chemical compound, drug | Palmitoyl alkyne-coenzyme A | Cayman chemical | Cat#15968 | |
| Chemical compound, drug | Alkyne palmitic acid | Click Chemistry Tools | Cat#1165–5 | |
| Software, algorithm | GraphPad9 | GraphPad | | |
| Other | TEAD luciferase reporter lentivirus | BPS Biosciences | Cat#79833 | Used to establish a stable cell line to monitor TEAD transcriptional activity; related to *Figure 6—figure supplement 3* (a) |
| Other | IRDye 800CW Streptavidin | LI-COR | Cat# 926–32230 | WB (0.2 µg/mL) |

## Cloning

The stretch of residues 209–424 of human TEAD1, residues 220–450 of human TEAD2, residues 119–436 of human TEAD3, residues 216–434 of human TEAD4, were inserted into the pET28PP (N-terminal His 3 C tag) vector.

## Protein expression and purification

The N-terminal His tag construct of human TEAD1 (residues 209–424) was overexpressed in *E. coli* BL21 (DE3) and purified using affinity chromatography and size-exclusion chromatography. Briefly, cells were grown at 37 °C in TB medium in the presence of 50 µg/ml of kanamycin to an OD of 0.8, cooled to 17 °C, induced with 500 µM isopropyl-1-thio-D-galactopyranoside (IPTG), incubated overnight at 17 °C, collected by centrifugation, and stored at –80 °C. Cell pellets were lysed in buffer A (25 mM HEPES, pH 7.5, 200 mM NaCl, 5% glycerol, 7 mM mercapto-ethanol, and 20 mM Imidazole) using Microfluidizer (Microfluidics), and the resulting lysate was centrifuged at 30,000 g for 40 min. Ni-NTA beads (Qiagen) were mixed with cleared lysate for 30 min and washed with buffer A. Beads were transferred to an FPLC-compatible column, and the bound protein was washed further with buffer A for 10 column volumes and eluted with buffer B (25 mM HEPES, pH 7.5, 200 mM NaCl, 5% glycerol, 7 mM mercapto-ethanol, and 400 mM Imidazole). The eluted sample was concentrated and purified further using a Superdex 200 16/600 column (Cytiva) in buffer C containing 20 mM HEPES, pH 7.5, 200 mM NaCl, 5% glycerol, 0.5 mM TCEP and 2 mM DTT. HRV 3 C protease was added to TEAD1 containing fractions and incubated overnight at 4 °C, followed by passing through Ni-NTA column to remove His-tag and 3 C protease. The flow-through fractions of the second Ni-NTA column, containing cleaved TEAD1, was concentrated to ~9 mg/mL and stored in –80 °C. The N-terminal His tag construct of human TEAD2 (residues 220–450), TEAD3 (residues 119–436) and TEAD4 (residues 216–434) were purified as TEAD1.

## Mass spectrometry analysis

TEAD2 protein (5 µg) was treated with DMSO or a 10-fold molar excess of MYF-03–69 and analyzed by LC-MS using an HPLC system (Shimadzu, Marlborough, MA) interfaced to an LTQ ion trap mass spectrometer (ThermoFisher Scientific, San Jose, CA). Proteins were desalted for 4 min on column with 100% A and eluted with an HPLC gradient (0–100% B in 1 min; A=0.2 M acetic acid in water; B=0.2 M acetic acid in acetonitrile). The mass spectrometer was programmed to acquire full scan mass spectra (*m/z* 300–2000) in profile mode (spray voltage = 4.5 kV). Mass spectra were deconvoluted using MagTran software version 1.03b2 *Zhang and Marshall, 1998*. To analyze the site of modification, DMSO or MYF-03–69 treated proteins were first captured on SP3 beads *Hughes et al., 2014* by adding an equal volume of acetonitrile and washed (2 x with 70% acetonitrile, 1 x with 100% acetonitrile). Beads were then resuspended with 100 mM ammonium bicarbonate containing 0.1% Rapigest (Waters, Milford, MA). Proteins were reduced with 10 mM DTT for 30 min at 56 °C, alkylated with 22.5 mM iodoacetamide for 30 min at room temperature, and then digested overnight with trypsin at

37 °C. Rapigest was cleaved according to the manufacturer's instructions, and peptides were desalted by C18 and dried by vacuum centrifugation. Peptides were reconstituted in 50% acetonitrile, 1% formic acid, 100 mM ammonium acetate, and analyzed by CE-MS using a ZipChip CE-MS instrument and autosampler (908 devices, Boston, MA) interfaced to a QExactive HF mass spectrometer (ThermoFisher Scientific). Peptides were resolved at 500 V/cm using an HR chip with a background electrolyte consisting of 50% acetonitrile with 1% formic acid. The mass spectrometer was operated in data dependent mode and subjected the 5 most abundant ions in each MS scan (*m/z* 300–2000, 60 K resolution, 3E6 target, 100ms max fill time) to MS/MS (15 K resolution, 1E5 target, 100ms max fill time). Dynamic exclusion was enabled with a repeat count of 1 and an exclusion duration of 5 s. Raw mass spectrometry data files were converted to.mgf using multiplierz software *Alexander et al., 2017* and searched against a forward-reverse human refseq database using Mascot version 2.6.2. Search parameters specified fixed carbamidomethylation of cysteine, variable methionine oxidation, and variable MYF-03–69 modification of cysteine. MYF-03–69 modified spectra were examined and figures prepared using mzStudio software *Ficarro et al., 2017*. Inhibitor related ions in MS/MS spectra were identified as described *Ficarro et al., 2016*.

## Docking

Docking of MYF-01–37 and MYF-03–69 was performed with covalent docking protocol from Schrodinger suite software (release 2019–02) with default parameters in TEAD2 structure (PDB code: 5HGU and 5DQ8, respectively) exporting 5 poses per molecule. Both stereoisomers were docked. Top scoring pose was chosen to illustrate the binding pose. The pdb structure was processed and optimized with protein preparation protocol with default setting in Schrodinger suite.

## Crystallization

Using Formulatrix NT8 and RockImager and ArtRobbins Phoenix liquid handlers, a 100 nL sample of 300 μM TEAD1 that was preincubated for 1 hr with 600 μM MYF-03–69 was dispensed in an equal volume of crystallization buffer (3 M $NH_4SO_4$ and 0.1 M Tris pH 9.0) and incubated against 25 μL of crystallization buffer in a 384-well hanging-drop vapor diffusion microtiter plate at 20 °C for 3 days.

## Data collection and structure determination

Diffraction data were collected at beamline 24ID-E of the NE-CAT at the Advanced Photon Source (Argonne National Laboratory). Data sets were integrated and scaled using XDS *Kabsch, 2010*. Structures were solved by molecular replacement using the program Phaser *McCoy et al., 2007* and available search models from the PDB. Iterative manual model building and refinement using Phenix *Adams et al., 2010* and Coot *Emsley and Cowtan, 2004* led to a model with excellent statistics.

## Gel-based palmitoylation studies

1 μM TEADs-YBD recombinant protein was incubated with inhibitors at the indicated concentrations at 37 °C for 2 hr followed by the addition of palmitoyl alkyne-coenzyme A (Cayman chemical, no. 15968) in a total volume of 50 μL. After 30 min reaction, 5 μL 10%SDS were added and 5 μL click reagents were added to start click reaction as previously reported . After another 1 hr, 4 x loading buffer were added to the reaction mixture and the samples subjected for western blot analysis. IRDye 800CW Streptavidin (LI-COR, no. 92632230) and His-Tag Mouse mAb (Cell Signaling, no. 2366 S) was used for biotin detection and His-tag detection. The blots were imaged on Odyssey CLx Imager (LI-COR).

## Sample preparation for SLC-ABPP

Samples for whole cysteine profiling were prepared as previously described . Briefly, frozen cell pellets from H226 cells were lysed using PBS (pH 7.4). Samples were further homogenized, and DNA was sheared using sonication with a probe sonicator (20x0.5 s pulses) at 4 °C. Total protein was determined using a BCA assay and cell lysates were used immediately for each experiment. Depending on the experiment, 50 μg of total cell extract was aliquoted for each TMT channel for further downstream processing. Excess DBIA, along with disulfide bonds were quenched and reduced using 5 mM dithiothreitol (DTT) for 30 min in the dark at room temperature. Subsequently reduced cysteine residues were alkylated using 20 mM iodoacetamide for 45 min in the dark at room temperature. To facilitate

removal of quenched DBIA and incompatible reagents, proteins were precipitated using chloroform/methanol. Briefly, to 100 µL of each sample, 400 µL of methanol were added, followed by 100 µL of chloroform with thorough vortexing. Next, 300 µL of HPLC grade water were added, and the samples were mixed to facilitate precipitation. Samples were centrifuged at maximum speed (14,000 rpm) for 3 min at room temperature, the aqueous top layer was removed, and the samples were washed additionally three times with 500 µL of methanol. Protein pellets were re-solubilized in 200 mM 4-(2-hydroxyethyl)–1-piperazinepropanesulfonic acid (EPPS) at pH 8.5 and digested using LysC and trypsin (1:100, enzyme to protein ratios) overnight at 37 °C using a ThermoMixer set to 1200 rpm. The next day samples were labeled with TMT reagents or stored at –80 °C until further use.

## TMT labeling for SLC-ABPP

Digested peptides containing DBIA conjugated-cysteines were labeled using TMTPro 16-plex reagents as previously described (**Navarrete-Perea et al., 2018**)[45a, 45b]. Briefly, peptides were labeled at a 1:2 ratio by mass (peptides to TMT reagents) for 1 hr with shaking at 1200 rpm. To equalize protein loading, ~2 µg of each sample were aliquoted, and a 60-min quality control analysis (ratio check) was performed using SPS-MS3. Excess TMT reagent was quenched with hydroxylamine (0.3% final concentration) for 15 min at room temperature. Next, samples were mixed 1:1 across all TMT channels and the pooled sample was dried using a Speedvac to ensure all acetonitrile was removed.

## Cysteine peptide enrichment using streptavidin in SLC-ABPP

Pierce streptavidin magnetic beads were washed with PBS pH 7.4 prior to use. To each TMT labeled pooled sample, 100 µL of a 50% slurry of streptavidin beads were added. Samples were further topped up with 1 mL of PBS in a 2 mL Eppendorf tube. Samples and beads were incubated overnight at 4 °C to enrich for TMT-labeled DBIA conjugated cysteine peptides. Following enrichment, the beads were placed onto a magnetic rack and allowed to equilibrate for 5 min. Beads were washed to remove non-specific binding using the following procedure: 3x1 mL of PBS pH 7.4, 1x1 mL of PBS with 0.1% SDS pH 7.4 and finally 3x1 mL HPLC grade water. Beads were resuspended using a pipette between washes and placed on the magnet between each wash. To elute cysteine containing peptides, 500 µL of 50% acetonitrile with 0.1% trifluoracetic acid (TFA) were added, and the beads were mixed at 1000 rpm for 15 min at room temperature. Eluted peptides were transferred to a new tube, and the beads were additionally washed with 200 µL of 50% acetonitrile with 0.1% TFA and combined. Cysteine containing peptides were dried to completion using a Speedvac and were stored at –80 °C.

## Desalting cysteine-containing peptides in SLC-ABPP

TMT-labeled cysteine-containing peptides were resuspended using 200 µL of 1% formic acid (FA) and were desalted using StageTips as previously described (**Navarrete-Perea et al., 2018**)[45a]. Briefly, eight 18-guage cores were packed into a 200-µL pipette tip and were passivated and equilibrated using the following solutions: 100 µL of 100% methanol, 70% acetonitrile 1% FA, and 5% acetonitrile 5% FA. Peptides were loaded and were washed with 0.1% FA and eluted using 150 µL of 70% acetonitrile 1% FA and dried to completion using a Speedvac. Enriched peptides samples were next resuspended in 5–10 µL of 5% acetonitrile 5% FA and between 50 and 100% of the sample was injected for analysis using LC-RTS-SPS-MS3.

## Mass spectrometry and real-time searching in SLC-ABPP

All SLC-ABPP mass spectrometry data were acquired using an Orbitrap Fusion Eclipse mass spectrometer in-line with a Proxeon NanoLC-1200 UPLC system. Peptides were separated using an in-house 100 µm capillary column packed with 35 cm of Accucore 150 resin (2.6 µm, 150 Å) (ThermoFisher Scientific) using 210 min gradients from 4 to 24% acetonitrile in 0.125% formic acid per run. Eluted peptides were quantified using the synchronous precursor selection (SPS-MS3) method for TMT quantification. Briefly, MS1 spectra were acquired at 120 K resolving power with a maximum of 50ms ion injection in the Orbitrap. MS2 spectra were acquired by selecting the top 10 most abundant features via collisional induced dissociation (CID) in the ion trap using an automatic gain control (AGC) setting of 15 K, quadrupole isolation width of 0.5 m/z and a maximum ion accumulation time of 50ms. These spectra were passed in real time to the external computer for online database searching. Intelligent data acquisition (IDA) using real-time searching (RTS) was performed using

Obiter *Schweppe et al., 2020*. Cysteine-containing peptides or whole proteome peptide spectral matches were analyzed using the Comet search algorithm (release_2019010) designed for spectral acquisition speed *Eng et al., 2015*. The same forward- and reversed-sequence human protein databases were used for both the RTS search and the final search (Uniprot). The RTS Comet functionality has been released and is available here: http://cometms.sourceforge.net/. Real-time access to spectral data was enabled by the Thermo Scientific Fusion API (https://github.com/thermofisherlsms/iapi; *Bailey and Canterbury, 2022*). Next, peptides were filtered using simple initial filters that included the following: not a match to a reversed sequence, maximum PPM error of <50, minimum XCorr of 0.5, minimum deltaCorr of 0.10 and minimum peptide length of 7. If peptide spectra matched to above criteria, an SPS-MS3 scan was performed using up to 10 *b-* and *y-type* fragment ions as precursors with an AGC of 200 K for a maximum of 200ms with a normalized collision energy setting of 55 (TMTPro 16-plex)[45b].

## Mass spectrometry data analysis in SLC-ABPP

All acquired data were searched using the open-source Comet algorithm (release_2019010) using a previously described informatics pipeline *McAlister et al., 2012*. Spectral searches were done using a custom FASTA-formatted database which included common contaminants, reversed sequences (Uniprot Human, 2014) and the following parameters: 50 PPM precursor tolerance, fully tryptic peptides, fragment ion tolerance of 0.9 Da and a static modification by TMTPro (+304.2071 Da) on lysine and peptide N termini. Carbamidomethylation of cysteine residues (+57.021 Da) was set as a static modification while oxidation of methionine residues (+15.995 Da) and DBIA on cysteine residues (+239.262) was set as a variable modification. Peptide spectral matches were filtered to a peptide false discovery rate (FDR) of less than 1% using linear discriminant analysis employing a target-decoy strategy. Resulting peptides were further filtered to obtain a 1% protein FDR at the entire dataset level (including all plexes per cell line), and proteins were collapsed into groups. Cysteine-modified peptides were further filtered for site localization using the AScore algorithm with a cutoff of 13 (p<0.05) as previously described[48b]. Overlapping peptide sequences that were generated from different charge states, elution times and tryptic termini were grouped together into a single entry. A single quantitative value was reported, and only unique peptides were reported. Reporter ion intensities were adjusted to correct for impurities during synthesis of different TMT reagents according to the manufacturer's specifications. For quantification of each MS3 spectrum, a total sum signal-to-noise of all reporter ions of 160 (TMTPro) was required. Lastly, peptide quantitative values were normalized so that the sum of the signal for all proteins in each channel was equal to account for sample loading differences (column normalization).

## SLC-ABPP competition ratio calculation

Cysteine site-specific engagement was assessed by the blockage of DBIA probe labeling. Peptides that showed >95% reduction in TMT intensities in the electrophile-treated samples were assigned a maximum ratio of 20 for graphing purposes with preserved ranking. TMT reporter ion sum-signal-to-noise for each SLC-ABPP experiment was used to calculate the competition ratios by dividing the control channel (DMSO) by the electrophile treated channel. Replicate measurements were averaged and reported as a single entry. To avoid false positives, sites with large coefficients of variation had the highest replicate CR values removed before averaging as previously described ; *Backus et al., 2016*.

## Cell culture

HEK293T (ATCC, no. CRL-3216), NCI-H226 (ATCC, no. CRL-5826), MSTO-211H (ATCC, no. CRL-2081), NCI-H2452 (ATCC, no. CRL-5946) mesothelioma cells, MeT-5A (ATCC, no. CRL-9444) mesothelium cells, 94T778 cells (ATCC, no. CRL-3044), 93T449 cells (ATCC, no. 3043), MIA PaCa-2 cells (ATCC, no. CRM-CRL-1420), MM.1S (ATCC, no. CRL-2974) and SKHEP-1 cells (ATCC, no. HTB-52) were obtained from American Type Culture Collection and cultured as recommended. HuCCT1 cells were gifts from Bardeesy lab in Massachusetts General Hospital. PC9 cells were gifts from Pasi lab in Dana-Farber Cancer Institute. Cells were negative for mycoplasma using MycoAlert mycoplasma detection kit (LONZA, no. LT07-418).

## Cell proliferation assay

For 2D adherent cell viability experiment, the cells were seeded at 384-well plate (Corning, no. 3570) at the density of 200 cells/well. The next day, compounds were added using Janus workstation (PerkinElmer). After 5 days treatment, the cell viability was measured by CellTiter-Glo kit (Promega, no. G7570) as the manufacturer recommended. For 3D spheroid assays, NCI-H226 and MSTO-211H cells were plated at the density of 200 cells/well in Ultra-low Attachment (ULA) plate (S-bio, no. MS-9384WZ) without or with 5% Matrigel matrix (Corning, no. 356231) respectively. The cell viability was measured using 3D CellTiter-Glo kit (Promega, no. G9681). The luminescent signal was collected on EnVision plate reader (PerkinElmer). The $GR_{50}$ values were calculated as previously described *Hafner et al., 2016*.

## Cell cycle analysis

Cells were plated in 6-well plate (Corning, no. 3506) and treated by MYF-03–69 at indicated concentrations. Then cells were harvested and fixed in cold 70% ethanol overnight. Next day, the samples were treated with 100 µg/mL RNAase A (Life Technologies, no. EN0531) and stained by 50 µg/mL propidium iodide solution (Life Technologies, no. P3566). After incubation at room temperature for 30 min, the samples were subjected to assessment using Guava flow cytometer. The data were further analyzed in Flowjo software.

## PRISM screening and data analysis

Up to 931 barcoded cell lines in pools of 20–25 were thawed and plated into 384-well plates (1250 cells/well for adherent cell pools, 2000 cells/well for suspension or mixed suspension/adherent cell pools) containing compound (top concentration: 10 µM, 8-point, threefold dilution). All conditions were tested in triplicate. Cells were lysed after 5 days of treatment and mRNA based Luminex detection of barcode abundance from lysates was carried out as described previously (*Corsello et al., 2020*). Luminex median fluorescence intensity (MFI) data was input to a standardized R pipeline (https://github.com/broadinstitute/prism_data_processing; *Boghossian, 2022*; copy archived at swh:1:rev:0b9f93140ed9b297c128fa0133acbea45dba19e6) to generate viability estimates relative to vehicle treatment for each cell line and treatment condition, and to fit dose-response curves from viability data. Correlation analysis was also performed in the R pipeline mentioned above.

## TEAD reporter assays

The TEAD transcriptional vector (TBS-mCherry) was used to produce lentivirus as previously reported in HEK293T cells *Mohseni et al., 2014*. The virus was collected at 48 hr and 72 hr post transfection and concentrated using Lenti-X concentrator (Takara Bio, no. 631231). Then NCI-H226 cells were transduced using the concentrated virus in the presence of 8 µg/mL polybrene (MilliporeSigma, no. TR1003G). The positively transduced cells were further sorted by GFP. For reporter assays, the selected cells were plated in black 384-well plate (Corning, no. 4514) at the density of 1000 cells/well. The next day, compounds were added at indicated concentrations using Janus workstation. After 72 hr incubation, the cells were stained by Hoechst 33342 (Life Technologies, no. 62249). The mCherry and Hoechst signals were read using Acumen high content imager (TTP Labtech).

For the luciferase reporter assay, NCI-H226 cells were transduced with 250 µL TEAD luciferase reporter lentivirus (BPS Biosciences, no. 79833) in the presence of 8 µg/mL polybrene (MilliporeSigma, no. TR1003G). Cells were then selected by 2 µg/mL puromycin. The selected cells were seeded at the density of 1000 cells/well and treated with compounds at indicated concentrations. After three days treatment, ONE-Glo luciferase assay kit (Promega, no. E6110) and CellTiter-Glo kit (Promega, no. G9681) were used according to manufacturer instructions. The TEAD transcriptional activity was then calculated by normalizing luciferase signal normalized with relative cell viability.

## Lipid displacement assay

The HEK-293T cells were plated in six-well plate (Corning, no. 3506) and transfected with Myc-TEAD4 vector the next day. The Myc-TEAD4 was a gift from Kunliang Guan *Li et al., 2010* (Addgene plasmid # 24638; http://n2t.net/addgene:24638; RRID: Addgene_24638). Then transfected cells were treated with 50 µM alkyne palmitic acid (Click Chemistry Tools, cat. no. 1165–5) in the presence of TEAD inhibitors at indicated concentrations next day. After 24 hr incubations, cells were collected and lysed in

RIPA buffer with proteasome inhibitor. The cleared supernatant was then subjected to click reactions using same procedures in the gel-based palmitoylation assay. Palmitoylation levels were detected by immunoblot using streptavidin antibody (LI-COR, no. 92632230).

### RT-PCR studies

The cells were plated in six-well plate (Corning, no. 3506) and treated with compounds the next day. The total RNA was extracted using RNeasy Plus Mini Kit (Qiagen, no. 74134). Then 500 ng purified RNA was used to synthesize cDNA by SuperScript III First-Strand Synthesis System Kit (Life Technologies, no. 18080051). The following TaqMan probes were used for follow-up RT-PCR reactions: CTGF (Hs00170014_m1), CYR61 (Hs00155479_m1), GAPDH (Hs02786624_g1), BMF (Hs00372937_m1), IGFBP3 (Hs00181211_m1), KRT34 (Hs02569742_s1) and NPPB (Hs00173590_m1).

### Co-immunoprecipitation (Co-IP) studies

The cells were plated in 10 cm dish (Corning, no. 430293) and treated with compounds at indicated concentrations the next day. The Co-IP experiments were conducted using Dynabeads Co-Immunoprecipitation Kit (Life Technologies, no. 14321D) according to the manufacturer instructions. The following antibodies were used for protein detection: pan-TEAD (1:1000, Cell Signaling #13295 S); TEAD4 (1:1000, Abcam#ab58310), YAP (1:1000, Cell Signaling #14074 S) and β-actin (1:2000, Cell Signaling #3700 S).

### RNA-Seq study: Sample treatment

The NCI-H226 cells were plated in the 10 cm dish and treated with MYF-03–69 at indicated concentrations with biological triplicates for 6 hours next day. The RNA was extracted using RNeasy plus mini kit (Qiagen, cat no.74134) according to the manufacturer instructions.

### RNA-Seq study: Library preparation and sequencing

Libraries were prepared using Roche Kapa mRNA HyperPrep strand specific sample preparation kits from 200 ng of purified total RNA according to the manufacturer's protocol on a Beckman Coulter Biomek i7. The finished dsDNA libraries were quantified by Qubit fluorometer and Agilent TapeStation 4200. Uniquely dual indexed libraries were pooled in an equimolar ratio and shallowly sequenced on an Illumina MiSeq to further evaluate library quality and pool balance. The final pool was sequenced on an Illumina NovaSeq 6000 targeting 40 million 100 bp read pairs per library at the Dana-Farber Cancer Institute Molecular Biology Core Facilities.

### RNA-Seq study: Analysis

Sequenced reads were aligned to the UCSC hg19 reference genome assembly and gene counts were quantified using STAR (v2.7.3a) *Dobin et al., 2013*. Differential gene expression testing was performed by DESeq2 (v1.22.1) *Love et al., 2014*. RNAseq analysis was performed using the VIPER snakemake pipeline *Cornwell et al., 2018*. KEGG pathway enrichment analysis was performed through metascape webportal .

### RNA-Seq study: Datasets availablility

These RNA-seq datasets were deposited to BioSample database under accession# as below:
SAMN19288936, SAMN19288937, SAMN19288938, SAMN19288939, SAMN19288940, SAMN19288941, SAMN19288942, SAMN19288943, SAMN19288944, SAMN19288945, SAMN19288946.

### Efficacy study in NCI-H226 CDX model: Animals

Female 7-week-old NSG mice were purchased from The Jackson Laboratory (Bar Harbor, ME). Animals acclimated for at least 5 days before initiation of the study. All in vivo studies were conducted at Dana-Farber Cancer Institute with the approval of the Institutional Animal Care and Use Committee in an AAALAC accredited vivarium.

## Efficacy study in NCI-H226 CDX model: In vivo studies

The NCI-H226 cells were grown in RPMI1640 media supplemented with 10% fetal bovine serum. Cells were harvested, and $5\times10^6$ cells with 50% Matrigel (Fisher Scientific) were implanted subcutaneously in the right flank of the NSG mice. Tumors were allowed to establish to an average of 141.3±24.9 mm³ in size before randomization using Studylog software (San Francisco, CA) into various treatment groups with 8–9 mice per group. MYF-03–176 was formulated as a suspension in 10% DMSO with 10% Tween 80 in water and dosed twice daily via oral gavage. Control treated mice received vehicle alone. Tumor volumes were determined from caliper measurements by using the formula, Tumor volume = (length × width *Zhao et al., 2008*)/2. Tumor volumes and body weights were measured twice weekly. Mice were treated for 28 days, followed by measuring for re-growth of tumors.

## Chemical synthesis and characterization
### MYF-03-42
Step 1: Synthesis of *trans*-tert-butyl 3-hydroxy-4-(1H-pyrazol-1-yl)pyrrolidine-1-carboxylate (Compound 3) (Scheme 1)

**Scheme 1.** Synthetic method of MYF-03-42.

To a suspension of 60% NaH (323 mg, 8.1 mmol) in THF (20 mL) was added the solution of 1H-pyrazole (551 mg, 8.1 mmol) in THF (5 mL), the mixture was stirred at 0 °C under $N_2$ for 30 min, and then tert-butyl 6-oxa-3-azabicyclo[3.1.0] hexane-3-carboxylate (1.0 g, 5.4 mmol) was added. The resulting mixture was heated at 60 °C under $N_2$ for 16 hr. After cooled down to room temperature the mixture was diluted with ethyl acetate (50 mL) and washed with water (50 mL). The organic layer was dried over anhydrous $Na_2SO_4$, filtered, concentrated and purified by flash column chromatography on silica gel (ethyl acetate in petroleum ether = 40% v/v) to afford compound **3** as solid (600 mg, yield 44%). LC-MS (ESI) m/z: 254 [M+H]⁺.

Step 2: Synthesis of *trans*-tert-butyl 3-(1H-pyrazol-1-yl)-4-(4-(trifluoromethyl) benzyloxy)pyrrolidine-1-carboxylate (Compound 5) (Scheme 1)

A mixture of compound **3** (300 mg, 1.18 mmol), 1-(bromomethyl)–4-(trifluoromethyl)benzene (283 mg, 1.18 mmol) and 60% NaH (47 mg, 1.18 mmol) in DMF (10 mL) was stirred at room temperature under $N_2$ for 2 hours. The mixture was diluted with water (50 mL) and extracted with ethyl acetate (50 mL x 2), the combined organic was washed with brine (50 mL), dried over anhydrous $Na_2SO_4$, filtered and concentrated to obtain compound **5** as yellow oil (400 mg, yield 82%). LC-MS (ESI) m/z: 356 [M-56+H]⁺.

Step 3: Synthesis of 1-(*trans*-4-(4-(trifluoromethyl)benzyloxy)pyrrolidin-3-yl)-1H-pyrazole (Compound 6) (Scheme 1)

A mixture of compound **5** (300 mg, 0.72 mmol) and TFA (3 mL) in DCM (10 mL) was stirred at room temperature for 2 hr. The mixture was concentrated to leave crude compound **6** (250 mg) as yellow oil, which was used directly in the next step. LC-MS (ESI) m/z: 312[M+H]⁺.

Step 4: Synthesis of 1-(*trans*-3-(1H-pyrazol-1-yl)-4-(4-(trifluoromethyl)benzyloxy)pyrrolidin-1-yl)prop-2-en-1-one (MYF-03-42) (Scheme 1)

A mixture of compound **6** (70 mg, 0.22 mmol), acryloyl chloride (20 mg, 0.22 mmol), and TEA (44 mg, 0.44 mmol) in DCM (10 mL) was stirred at room temperature under $N_2$ for 2 hr. The mixture was concentrated, the residue was purified by prep-HPLC to afford **MYF-03–42** as yellow oil (50 mg, yield 62%). LC-MS (ESI) m/z: 366[M+H]⁺.

¹H-NMR (400 MHz, CD₃OD): δ (ppm) 7.74 (dd, J=5.2, 2.4 Hz, 1 H), 7.63 (d, J=8.0 Hz, 2 H), 7.56 (d, J=1.6 Hz, 1 H), 7.47 (d, J=8.1 Hz, 2 H), 6.66–6.59 (dd, J=16.8, 10.4 Hz, 1 H), 6.36–6.30 (m, 2 H),

5.85–5.74 (m, 1 H), 5.18–5.02 (m, 1 H), 4.68 (d, J=5.8 Hz, 2 H), 4.51–4.42 (m, 1 H), 4.28–4.08 (m, 2 H), 4.01–3.93 (m, 1 H), 3.83–3.65 (m, 1 H).

## MYF-02-111

was synthesized through the same route as **MYF-03–42** except 1-(bromomethyl)–3-(trifluoromethyl) benzene was used in **Step 2** instead of 1-(bromomethyl)–4-(trifluoromethyl)benzene. **MYF-02–111** was obtained as light yellow oil. LC-MS (ESI) m/z: 366[M+H]⁺. ¹H NMR (500 MHz, DMSO-$d_6$) δ 7.87 (ddd, J=9.3, 2.3, 0.7 Hz, 1 H), 7.69–7.54 (m, 4 H), 7.53–7.48 (m, 1 H), 6.61 (dd, J=16.8, 10.3 Hz, 1 H), 6.30 (dt, J=3.4, 2.1 Hz, 1 H), 6.17 (dt, J=16.7, 2.5 Hz, 1 H), 5.70 (td, J=9.9, 2.3 Hz, 1 H), 5.13 (ddt, J=33.6, 7.8, 3.4 Hz, 1 H), 4.66 (d, J=4.0 Hz, 2 H), 4.39 (ddd, J=25.5, 4.1, 1.7 Hz, 1 H), 4.19–4.09 (m, 0.5 H), 4.07–3.87 (m, 1.5 H), 3.86–3.70 (m, 1.5 H), 3.52 (dd, J=13.0, 4.0 Hz, 0.5 H).

## MYF-03-69

### Step 1: Synthesis of (*3R, 4R*)-tert-butyl 3-azido-4-hydroxypyrrolidine-1-carboxylate (Compound 2) (Scheme 2)

**Scheme 2.** Synthetic method of MYF-03-69.

A mixture of tert-butyl 6-oxa-3-azabicyclo[3.1.0]hexane-3-carboxylate (4 g, 21.6 mmol), TMSN₃ (2.664 g, 23.2 mmol) and chiral catalyst (1 S,2S)-(-)-[1,2-cyclohexanediamino-N,N'-bis(3,5-di-t-butylsalicylidene)]chromium(III) chloride (328 mg, 0.42 mmol) was stirred at rt under N₂ overnight. The reaction mixture was treated with MeOH (60 mL) and K₂CO₃ (1.788 g, 12.8 mmol) and continued to stir at rt for 5 hours. The reaction mixture was diluted with ethyl acetate (300 mL), washed with water (300 mL x 2), dried over anhydrous Na₂SO₄, concentrated and purified by flash column chromatography on silica gel (ethyl acetate in petroleum ether = 30% v/v) to obtain the compound **2** as clear oil (3.5 g, 96% e.e., yield 71%). LC-MS (ESI) m/z: 129 [M+H-100]⁺.

### Step 2: Synthesis of (*3R, 4R*)-tert-butyl 3-azido-4-(4-(trifluoromethyl)benzyloxy)pyrrolidine-1-carboxylate (Compound 4) (Scheme 2)

A mixture of compound **2** (3 g, 13.1 mmol), 1-(bromomethyl)–4-(trifluoromethyl)benzene (3.1 g, 13.1 mmol) and 60% NaH (0.6 g, 15.7 mmol) in DMF (20 mL) was stirred at 0 °C under N₂ for 6 hours. The reaction mixture was diluted with water (200 mL) and extracted with ethyl acetate (200 mL), the organic was washed with water (100 mL), dried over anhydrous Na₂SO₄, concentrated and purified by flash column chromatography on silica gel (ethyl acetate in petroleum ether = 20% v/v) to obtain compound **4** as oil (3.8 g, yield 75%). LC-MS (ESI) m/z: 287[M+H-100]⁺.

### Step 3: Synthesis of (*3R, 4R*)-tert-butyl 3-(4-(pyridin-3-yl)-1H-1,2,3-triazol-1-yl)-4-(4-(trifluoromethyl)benzyloxy)pyrrolidine-1-carboxylate (Compound 6) (Scheme 2)

A mixture of compound **4** (3.8 g, 9.8 mmol), 3-ethynylpyridine (1.216 g, 11.8 mmol), CuSO₄ (244 mg, 0.98 mmol) and, sodium L-ascorbate (388 mg, 1.96 mmol) in THF (20 mL), H₂O (20 mL) and ⁿBuOH (20 mL) was stirred at 70 °C under N₂ overnight. The reaction mixture was concentrated in vacuum, the residue was diluted with ethyl acetate (600 mL), washed with water (400 mL), dried over anhydrous Na₂SO₄, concentrated and purified by flash column chromatography on silica gel (ethyl acetate in

petroleum ether = 90% v/v) to obtain compound **6** as yellow solid (4 g, yield 83%). LC-MS (ESI) m/z: 490[M+H]$^+$.

### Step 4: Synthesis of 3-(1-((*3R, 4R*)-4-(4-(trifluoromethyl)benzyloxy)pyrrolidin-3-yl)-1H-1,2,3-triazol-4-yl)pyridine (Compound 7) (Scheme 2)

A mixture of compound **6** (4 g, 8.1 mmol) and TFA (5 mL) in DCM (20 mL) was stirred at 0 °C under N$_2$ for 2 hours. The mixture was concentrated to leave the crude compound **7** (4 g, crude) as yellow oil, which was used directly for next step. LC-MS (ESI) m/z: 390[M+H]$^+$.

### Step 5: Synthesis of 1-((*3 R, 4 R*)–3-(4-(pyridin-3-yl)–1 H-1,2,3-triazol-1-yl)–4-(4-(trifluoromethyl)benzyloxy)pyrrolidin-1-yl)prop-2-en-1-one (MYF-03–69) (Scheme 2)

A mixture of compound **7** (500 mg, 1.28 mmol), acryloyl chloride (115 mg, 1.28 mmol) and TEA (388 mg, 3.84 mmol) in DCM (20 mL) was stirred at rt for 2 hours. The mixture was concentrated and purified by prep-HPLC to obtain **MYF-03–69** as light yellow solid (300 mg, yield 52%). LC-MS (ESI) m/z: 444 [M+H]$^+$. $^1$H NMR (400 MHz, CD$_3$OD) δ (ppm) 8.91 (d, *J*=1.8 Hz, 1 H), 8.50 (d, *J*=4.9 Hz, 1 H), 8.43 (d, *J*=4.9 Hz, 1 H), 8.18 (dt, *J*=8.0, 1.8 Hz, 1 H), 7.51 (d, *J*=8.0 Hz, 2 H), 7.48–7.38 (m, 3 H), 6.60–6.49 (m, 1 H), 6.23 (dd, *J*=16.8, 1.9 Hz, 1 H), 5.75–5.67 (m, 1 H), 5.41–5.30 (m, 1 H), 4.68 (d, *J*=4.0 Hz, 2 H), 4.60–4.49 (m, 1 H), 4.32–4.16 (m, 1 H), 4.15–3.87 (m, 2 H), 3.85–3.63 (m, 1 H). $^{13}$C NMR (126 MHz, DMSO-$d_6$) δ 161.46 (d, *J*=7.5 Hz), 159.37 (t, *J*=32.7 Hz), 158.06, 157.83, 155.71, 143.24, 128.17, 127.95 (d, *J*=9.2 Hz), 125.15 (q, *J*=4.5 Hz), 110.89 (d, *J*=5.7 Hz), 100.01 (dd, *J*=18.6, 15.4 Hz), 81.00 (d, *J*=3.5 Hz), 78.41, 69.27, 55.20 (d, *J*=2.8 Hz), 52.44, 51.26 (d, *J*=8.8 Hz), 50.85 (d, *J*=10.1 Hz), 50.60, 50.23. HRMS (ESI) calcd. for C22H21F3N5O2 (M+H)$^+$: 444.1642, Found: 444.1645.

## NC-MYF-03-69

### Synthesis of 1-((*3 R,4R*)–3-(4-(pyridin-3-yl)–1 H-1,2,3-triazol-1-yl)–4-((4-(tri-fluoromethyl)benzyl)oxy)pyrrolidin-1-yl)propan-1-one (NC-MYF-03–69) (Scheme 3)

**Scheme 3.** Synthetic method of NC-MYF-03-69.

A mixture of compound 7 (80 mg, 0.2 mmol, from the synthesis of **MYF-03–69**), propionyl chloride (18 mg, 0.2 mmol) and TEA (40 mg, 0.4 mmol) in DCM (5 mL) was stirred at room temperature under N$_2$ for 2 hours. The mixture was concentrated in vacuum, the residue was purified by prep-HPLC to obtain compound **NC-MYF-03–69** as white solid (37 mg, yield 42%). LC-MS (ESI) m/z: 446.3 [M+H]$^+$. $^1$H NMR (400 MHz, CD$_3$OD) δ (ppm) 9.03 (s, 1 H), 8.58 (dd, *J*=21.8, 7.7 Hz, 2 H), 8.39–8.19 (m, 1 H), 7.82–7.41 (m, 5 H), 5.67–5.28 (m, 1 H), 4.79 (d, *J*=6.4 Hz, 2 H), 4.72–4.57 (m, 1 H), 4.36–3.90 (m, 3 H), 3.76 (ddd, *J*=16.1, 12.3, 3.3 Hz, 1 H), 2.57–2.28 (m, 2 H), 1.16 (t, *J*=7.4 Hz, 3 H).

## MYF-03-162

### Step 1: Synthesis of compound 2' (Scheme 4)

**Scheme 4.** Synthetic method of MYF-03-162.

A mixture of tert-butyl 6-oxa-3-azabicyclo[3.1.0]hexane-3-carboxylate (10 g, 54 mmol), $NaN_3$ (7 g, 108 mmol), and $NH_4Cl$ (2.8 g, 54 mmol) in MeOH (120 mL) and $H_2O$ (20 mL) was stirred at 65 °C under $N_2$ overnight. The reaction mixture was concentrated in vacuum, the residue was extracted with ethyl acetate (300 mL x 3), the combined organic was washed with water (200 mL x2), dried over anhydrous $Na_2SO_4$, concentrated and purified by flash column chromatography on silica gel (ethyl acetate in petroleum ether = 20% v/v) to obtain compound **2'** (11 g, yield 89%) as oil. LC-MS (ESI) m/z: 129 [M+H-100]$^+$.

### Step 2: Synthesis of compound 4' (Scheme 4)

A mixture of compound **2'** (3 g, 13.1 mmol), 1-(bromomethyl)–4-(trifluoromethyl)benzene (3.1 g, 13.1 mmol) and 60% NaH (0.6 g, 15.7 mmol) in DMF (20 mL) was stirred at rt under $N_2$ protection for 6 hr. The reaction mixture was diluted with water (100 mL) and extracted with ethyl acetate (100 mL x 2), the combined organic was washed with water (100 mL), dried over anhydrous $Na_2SO_4$, concentrated and purified by flash column chromatography on silica gel (ethyl acetate in petroleum ether = 30% v/v) to obtain compound **4'** (3.8 g, yield 75%) as oil. LC-MS (ESI) m/z: 287 [M+H-100]$^+$.

### Step 3: Synthesis of 4-ethynylpiperidine (compound 2) (Scheme 4)

A mixture of tert-butyl 4-ethynylpiperidine-1-carboxylate (1000 mg, 4.7 mmol) and TFA (0.5 mL) in DCM (1 mL) was stirred at rt under $N_2$ protection for 2 hr. The mixture was concentrated to leave the crude compound **2** (1.2 g) as white solid, which was used directly in the next step. LC-MS (ESI) m/z: 110 [M+H]$^+$.

### Step 4: Synthesis of *trans*-tert-butyl 3-(4-(piperidin-4-yl)-1H-1,2,3-triazol-1-yl)-4-(4-(trifluoromethyl)benzyloxy)pyrrolidine-1-carboxylate (compound 3) (Scheme 4)

A mixture of compound **4'** (200 mg, 0.52 mmol), compound **2** (113 mg, 1.04 mmol), $CuSO_4$ (48 mg, 0.3 mmol) and sodium L-ascorbate (28 mg, 0.14 mmol) in THF (1 mL), $H_2O$ (1 mL) and $^nBuOH$ (1 mL) was stirred at 70 °C under $N_2$ overnight. The reaction mixture was concentrated in vacuum, the residue was diluted with water (50 mL) and extracted with ethyl acetate (20 mL x 2), the combined organic was washed with brine (20 mL), dried over anhydrous $Na_2SO_4$, concentrated and purified by flash column

chromatography on silica gel (ethyl acetate in petroleum ether = 90% v/v) to obtain compound **3** as yellow solid (200 mg, yield 78%). LC-MS (ESI) m/z: 496[M+H]⁺.

### Step 5: Synthesis of *trans*-tert-butyl 3-(4-(1-methylpiperidin-4-yl)-1H-1,2,3-triazol-1-yl)-4-(4-(trifluoromethyl)benzyloxy)pyrrolidine-1-carboxylate (compound 4) (Scheme 4)

A mixture of compound **3** (100 mg, 0.2 mmol) and formaldehyde (10 mg, 0.3 mmol) and NaBH₄ (20 mg, 0.4 mmol) in MeOH (10 mL) was stirred at rt for 2 hr. The resulting mixture was concentrated and purified by flash column chromatography on silica gel (ethyl acetate in petroleum ether = 80% v/v) to obtain compound **4** as oil (70 mg, yield 69%). LC-MS (ESI) m/z: 510[M+H]⁺.

### Step 6: Synthesis of 1-methyl-4-(1-(*trans*-4-(4-(trifluoromethyl)benzyloxy) pyrrolidin-3-yl)-1H-1,2,3-triazol-4-yl)piperidine (compound 5) (Scheme 4)

A mixture of compound **4** (70 mg, 0.13 mmol) and TFA (0.5 mL) in DCM (1 mL) was stirred at rt under N₂ for 2 hr. The mixture was concentrated to leave the crude compound **5** (50 mg) as yellow oil, which was used directly in the next step. LC-MS (ESI) m/z: 410[M+H]⁺.

### Step 7: Synthesis of 1-(*trans*-3-(4-(1-methylpiperidin-4-yl)—1 H-1,2,3-tri-azol-1-yl)—4-(4-(trifluoromethyl)benzyloxy)pyrrolidin-1-yl)prop-2-en-1-one (Compound MYF-03–162) (Scheme 4)

To the mixture of compound **5** (50 mg, 0.12 mmol) and TEA (30 mg, 0.24 mmol) in DCM (10 mL) was added acryloyl chloride (20 mg, 0.12 mmol), the mixture was stirred at rt under N₂ for 2 hours, and then concentrated and purified by prep-HPLC to obtain compound **MYF-03–162** as white solid (13 mg, yield 23%). LC-MS (ESI) m/z: 464 [M+H]⁺; ¹ H NMR (400 MHz, CD₃OD) δ (ppm) 7.89 (d, J=3.6 Hz, 1 H), 7.63 (d, J=8.1 Hz, 2 H), 7.49(d, J=8.1 Hz, 2 H), 6.61 (dd, J=16.8, 10.4 Hz, 1 H), 6.31 (dd, J=16.8, 1.8 Hz, 1 H), 5.78 (ddd, J=10.4, 3.6, 1.9 Hz, 1 H), 5.39–5.27 (m, 1 H), 4.73 (d, J=4.6 Hz, 2 H), 4.61–4.50 (m, 1 H), 4.34–4.10 (m, 2 H), 4.09–3.69 (m, 2 H), 2.94 (d, J=11.6 Hz, 2 H), 2.80–2.68 (m, 1 H), 2.31 (s, 3 H), 2.24–2.12 (m, 2 H), 2.08–1.98 (m, 2 H), 1.80–1.65 (m, 2 H).

## MYF-03-139

### Step 1: Synthesis of *trans*-tert-butyl 3-hydroxy-4-(4-(pyridin-3-yl)-1H-1,2,3-tri-azol-1-yl)pyrrolidine-1-carboxylate (compound 3') (Scheme 5)

**Scheme 5.** Synthetic method of MYF-03-139.

A mixture of tert-butyl compound **1'** (1 g, 4.38 mmol), 3-ethynylpyridine (451 mg, 4.38 mmol), CuSO₄ (654 mg, 2.6 mmol) and sodium L-ascorbate (257 mg, 1.3 mmol) in THF (3 mL), H₂O (3 mL) and

$^n$BuOH (3 mL) was stirred at 70 °C under $N_2$ protection overnight. The reaction mixture was diluted with ethyl acetate (60 mL), washed with water (40 mL), dried over anhydrous $Na_2SO_4$, concentrated and purified by flash column chromatography on silica gel (ethyl acetate in petroleum ether = 80% v/v) to obtain compound **3'** (650 mg, yield 45%) as yellow solid. LC-MS (ESI) m/z: 332[M+H]$^+$.

### Step 2: Synthesis of (4,4-difluorocyclohexyl)methyl trifluoromethanesulfonate (compound 2) (Scheme 5)

A mixture of (4,4-difluorocyclohexyl)methanol (1 g, 6.6 mmol), Tf$_2$O (2.81 g, 9.9 mmol) and pyridine (1 mL) in DCM (20 mL) was stirred at rt under $N_2$ protection for 3 hours. The reaction mixture was diluted with ethyl acetate (100 mL), washed with water (100 mL x 2), dried over anhydrous $Na_2SO_4$, filtered and concentrated to leave the crude compound **2** (500 mg) as yellow oil, which was used directly in the next step. LC-MS (ESI) m/z: no MS.

### Step 3: Synthesis of trans-tert-butyl 3-((4,4-difluorocyclohexyl)methoxy)-4-(4-(pyridin-3-yl)-1H-1,2,3-triazol-1-yl)pyrrolidine-1-carboxylate (compound 3) (Scheme 5)

A mixture of compound **2** (200 mg, 0.7 mmol), compound **3'** (58 mg, 0.175 mmol) and $^t$BuONa (25 mg, 0.26 mmol) in THF (5 mL) was stirred at 0 °C under $N_2$ for 2 hr. The reaction mixture was diluted with ethyl acetate (50 mL), washed with water (50 mL x 2), dried over anhydrous $Na_2SO_4$, concentrated and purified by flash column chromatography on silica gel (ethyl acetate in petroleum ether = 50% v/v) to obtain compound **3** as oil (100 mg, yield 30%). LC-MS (ESI) m/z: 464[M+H]$^+$.

### Step 4: Synthesis of 3-(1-(trans-4-((4,4-difluorocyclohexyl)methoxy)pyrrolidin-3-yl)-1H-1,2,3-triazol-4-yl)pyridine (compound 4) (Scheme 5)

A mixture of compound **3** (50 mg, 0.1 mmol) and TFA (1 mL) in DCM (3 mL) was stirred at rt for 2 hr. The mixture was concentrated to leave the crude compound **4** (50 mg) as yellow oil, which was used directly in the next step. LC-MS (ESI) m/z: 364[M+H]$^+$.

### Step 5: Synthesis of 1-(trans-3-((4,4-difluorocyclohexyl)methoxy)-4-(4-(pyridin-3-yl)-1H-1,2,3-triazol-1-yl)pyrrolidin-1-yl)prop-2-en-1-one (compound MYF-03-139) (Scheme 5)

A mixture of compound **4** (50 mg, 0.1 mmol), acryloyl chloride (10 mg, 0.1 mmol), and TEA (20 mg, 0.2 mmol) in DCM (5 mL) was stirred at rt under $N_2$ for 2 hr. The mixture was diluted with DCM (50 mL), washed with water (50 mL x 2), dried over anhydrous $Na_2SO_4$, concentrated and purified by prep-HPLC to obtain **MYF-03–139** as white solid (16 mg, yield 38%). LC-MS (ESI) m/z: 418[M+H]$^+$; $^1$H-NMR (400 MHz, CD$_3$OD) δ (ppm) 8.91 (d, J=1.2 Hz, 1 H), 8.55–8.40 (m, 2 H), 8.18 (dt, J=8.0, 1.9 Hz, 1 H), 7.43 (dd, J=8.0, 4.9 Hz, 1 H), 6.55 (ddd, J=16.8, 10.4, 2.5 Hz, 1 H), 6.28–6.20 (m, 1 H), 5.70 (ddd, J=10.5, 4.7, 1.9 Hz, 1 H), 5.29–5.18 (m, 1 H), 4.43–4.32 (m, 1 H), 4.27–4.16 (m, 1 H), 4.07–3.37 (m, 5 H), 1.91 (ddd, J=13.9, 7.0, 3.5 Hz, 2 H), 1.76–1.54 (m, 5 H), 1.26–1.13 (m, 2 H).

## MYF-03-137

### Step 1: Synthesis of (6-(trifluoromethyl)pyridin-3-yl)methanol (compound 2) (Scheme 6)

**Scheme 6.** Synthetic method of MYF-03-137.

A mixture of 6-(trifluoromethyl)nicotinaldehyde (170 mg, 1 mmol) and NaBH$_4$ (76 mg, 2 mmol) in MeOH (3 mL) was stirred at 0 °C for 3 hr. The reaction mixture was concentrated in vacuum, the residue was extracted with ethyl acetate (60 mL), washed with water (40 mL), dried over anhydrous Na$_2$SO$_4$, filtered and concentrated to leave the crude product (150 mg, yield 84%) as oil, which was used directly in the next step. LC-MS (ESI) m/z: 178[M+H]$^+$.

### Step 2: Synthesis of 5-(bromomethyl)-2-(trifluoromethyl)pyridine (Compound 3) (Scheme 6)

A mixture of compound **2** (400 mg, 2.2 mmol) and 48% aqueous HBr solution (6 mL) was stirred at 110 °C overnight. The reaction mixture was concentrated in vacuum, the residue was diluted with ethyl acetate (60 mL), washed with water (40 mL), dried over anhydrous Na$_2$SO$_4$, concentrated and purified by flash column chromatography on silica gel (ethyl acetate in petroleum ether = 20% v/v) to obtain compound **3** as oil (200 mg, yield 38%). LC-MS (ESI) m/z: 240[M+H]$^+$.

### Step 3: Synthesis of *trans*-tert-butyl 3-(4-(pyridin-3-yl)-1H-1,2,3-triazol-1-yl)-4-((6-(trifluoromethyl)pyridin-3-yl)methoxy)pyrrolidine-1-carboxylate (compound 5) (Scheme 6)

A mixture of compound **3** (100 mg, 0.4 mmol), compound **4** (140 mg, 0.4 mmol) and 60% NaH (32 mg, 0.8 mmol) in DMF (5 mL) was stirred at rt under N$_2$ overnight. The reaction mixture was diluted with water (100 mL) and extracted with ethyl acetate (20 mL x 2), the combined organic was washed with water (50 mL), dried over anhydrous Na$_2$SO$_4$, concentrated and purified by flash column chromatography on silica gel (ethyl acetate in petroleum ether = 90% v/v) to obtain compound **5** as solid (100 mg, yield 51%). LC-MS (ESI) m/z: 491[M+H]$^+$.

### Step 4: Synthesis of 5-((*trans*-4-(4-(pyridin-3-yl)-1H-1,2,3-triazol-1-yl)pyrrolidin-3-yloxy)methyl)-2-(trifluoromethyl)pyridine (compound 6) (Scheme 6)

A mixture of compound **5** (100 mg, 0.2 mmol) and TFA (1 mL) in DCM (3 mL) was stirred at rt for 2 hr. The mixture was concentrated to leave the crude compound **6** (100 mg, crude) as yellow oil, which was used directly in the next step. LC-MS (ESI) m/z: 391.

### Step 5: Synthesis of 1-(*trans*-3-(4-(pyridin-3-yl)–1 H-1,2,3-triazol-1-yl)–4-((6-(trifluoromethyl)pyridin-3-yl)methoxy)pyrrolidin-1-yl)prop-2-en-1-one (compound MYF-03–137) (Scheme 6)

A mixture of compound **6** (50 mg, 0.125 mmol), acryloyl chloride (15 mg, 0.125 mmol) and TEA (25 mg, 0.25 mmol) in DCM (3 mL) was stirred at 0 °C for 2 hr. The mixture was concentrated and purified by prep-HPLC to obtain **MYF-03–137** as white solid (5 mg, yield 9%). LC-MS (ESI) m/z: 445. $^1$H NMR (400 MHz, CD$_3$OD) δ (ppm) 9.04 (d, *J*=2.2 Hz, 1 H), 8.76–8.50 (m, 3 H), 8.31 (dt, *J*=7.9, 1.9 Hz,

1 H), 8.03 (d, J=8.2 Hz, 1 H), 7.80 (d, J=8.1 Hz, 1 H), 7.56 (dd, J=8.0, 4.9 Hz, 1 H), 6.67 (ddd, J=16.8, 10.4, 3.5 Hz, 1 H), 6.36 (dd, J=16.8, 1.7 Hz, 1 H), 5.83 (ddd, J=10.5, 5.1, 1.9 Hz, 1 H), 5.58–5.44 (m, 1 H), 4.97 (s, 2 H), 4.81–4.68 (m, 1 H), 4.48–4.03 (m, 3 H), 4.02–3.80 (m, 1 H).

## MYF-03-138

### Step 1: Synthesis of 4-((trimethylsilyl)ethynyl)pyridin-2(1H)-one (compound 3) (Scheme 7)

**Scheme 7.** Synthetic method of MYF-03-138.

To the solution of ethynyltrimethylsilane (0.595 g, 6.07 mmol) in DMF (40 mL) was added 4-bromopyridin-2(1 H)-one (1 g, 5.8 mmol), PdCl$_2$(PPh$_3$)$_2$ (0.204 g, 0.29 mmol), CuI (55 mg, 0.29 mmol) and Et$_3$N (1.17 g, 11.6 mmol). The mixture was stirred at 90 °C under N$_2$ for 2 hr. After cooled down to rt the mixture was diluted with water (200 mL) and extracted with EtOAc (50 mL x 2), the combined organic was dried over anhydrous Na$_2$SO$_4$, concentrated and purified by flash column chromatography on silica gel (ethyl acetate in petroleum ether = 50% v/v) to obtain compound **3** as oil (500 mg, yield 45.4%). LC-MS (ESI) m/z: 192[M+H]$^+$.

### Step 2: Synthesis of 4-ethynylpyridin-2(1H)-one (compound 4) (Scheme 7)

To the solution of compound **3** (450 mg, 2.35 mmol) in MeOH (20 mL) was added KOH (263 mg, 4.70 mmol). The mixture was stirred at rt under N$_2$ for 2 hr. The resulted mixture was concentrated and purified by flash column chromatography on silica gel (ethyl acetate in petroleum ether = 90% v/v) to obtain compound **4** as oil (200 mg, yield 71.4%). LC-MS (ESI) m/z: 120[M+H]$^+$.

### Step 3: Synthesis of *trans*-tert-butyl 3-(4-(2-oxo-1,2-dihydropyridin-4-yl)-1H-1,2,3-triazol-1-yl)-4-(4-(trifluoromethyl)benzyloxy)pyrrolidine-1-carboxylate (compound 6) (Scheme 7)

To the solution of compound **5** (250 mg, 0.65 mmol) in THF (10 mL), H$_2$O (10 mL) and $^n$BuOH (10 mL) was added compound **4** (116 mg, 0.97 mmol), CuSO$_4$ (15 mg, 0.065 mmol) and sodium L-ascorbate (26 mg, 0.13 mmol). The mixture was stirred at 70 °C under N$_2$ for 16 hr. The resulting mixture was concentrated and purified by flash column chromatography on silica gel (methanol in dichloromethane = 20% v/v) to obtain compound **6** as solid (200 mg, yield 60%). LC-MS (ESI) m/z: 506[M+H]+.

### Step 4: Synthesis of 4-(1-(*trans*-4-(4-(trifluoromethyl)benzyloxy)pyrrolidin-3-yl)-1H-1,2,3-triazol-4-yl)pyridin-2(1H)-one (compound 7) (Scheme 7)

To the solution of compound **6** (180 mg, 0.36 mmol) in DCM (10 mL) was added TFA (2 mL). The mixture was stirred at rt for 2 hours and concentrated in vacuum, the residue was adjusted to pH~8 with NaHCO$_3$ solution and extracted with EtOAc (50 mL x 3), the combined organics were washed with brine (100 mL), dried over anhydrous Na$_2$SO$_4$, filtered and concentrated to leave crude compound **7** as oil (150 mg, crude). LC-MS (ESI) m/z: 406[M+H]$^+$.

### Step 5: Synthesis of 4-(1-(*trans*-1-acryloyl-4-(4-(trifluoromethyl)benzyloxy) pyrrolidin-3-yl)-1H-1,2,3-triazol-4-yl)pyridin-2(1H)-one (compound MYF-03-138) (Scheme 7)

To the solution of compound **7** (130 mg, 0.32 mmol) in THF (10 mL) was added acryloyl chloride (29 mg, 0.32 mmol) and Et₃N (65 mg, 0.64 mmol). The mixture was stirred at 0 °C for 1 hour, and then concentrated and purified by prep-HPLC to obtain compound **MYF-03–138** as solid (44 mg, yield 30.0%). LC-MS (ESI) m/z: 460[M+H]⁺. ¹H-NMR (400 MHz, DMSO-*d₆*) δ (ppm) 11.58 (br, 1 H), 8.89 (d, *J*=5.5 Hz, 1 H), 7.70 (d, *J*=8.0 Hz, 2 H), 7.54 (d, *J*=8.0 Hz, 2 H), 7.46 (d, *J*=6.6 Hz, 1 H), 6.79 (s, 1 H), 6.70–6.56 (m, 2 H), 6.24–6.14 (m, 1 H), 5.79–5.71 (m, 1 H), 5.53–5.39 (m, 1 H), 4.81–4.71 (m, 2 H), 4.62–4.50 (m, 1 H), 4.30–4.13 (m, 1 H), 4.12–3.95 (m, 1 H), 3.91–3.79 (m, 1 H), 3.68–3.58 (m, 1 H).

## MYF-03-146

### Step 1: Synthesis of 1-methyl-4-((trimethylsilyl)ethynyl)pyridin-2(1H)-one (compound 3) (Scheme 8)

**Scheme 8.** Synthetic method of MYF-03-146.

To the solution of 4-bromo-1-methylpyridin-2(1 H)-one (1 g, 5.3 mmol) in DMF (20 mL) was added ethynyltrimethylsilane (0.55 g, 5.6 mmol), PdCl₂(PPh₃)₂ (0.21 g, 0.3 mmol), CuI (0.06 g, 0.3 mmol) and Et₃N (1.07 g, 10.6 mmol). The mixture was stirred at room temperature under N₂ for 2 hr. The resulting mixture was concentrated and purified by flash column chromatography on silica gel (ethyl acetate in petroleum ether = 50% v/v) to obtain compound **3** as oil (1.1 g, yield 99.9%). LC-MS (ESI) m/z: 206[M+H]⁺.

### Step 2: Synthesis of 4-ethynyl-1-methylpyridin-2(1H)-one (compound 4) (Scheme 8)

To the solution of compound **3** (1 g, 4.9 mmol) in MeOH (20 mL) was added K₂CO₃ (1.35 g, 9.8 mmol). The mixture was stirred at room temperature under N₂ for 2 hr. The resulting mixture was concentrated and purified by flash column chromatography on silica gel (ethyl acetate in petroleum ether = 90% v/v) to obtain compound **4** as solid (400 mg, yield 56.1%). LC-MS (ESI) m/z: 134[M+H]⁺.

### Step 3: Synthesis of *trans*-tert-butyl 3-(4-(1-methyl-2-oxo-1,2-dihydropyridin-4-yl)-1H-1,2,3-triazol-1-yl)-4-(4-(trifluoromethyl)benzyloxy)pyrrolidine-1-carboxylate (compound 6) (Scheme 8)

To the solution of *trans*-tert-butyl 3-azido-4-(4-(trifluoromethyl)benzyloxy) pyrrolidine-1-carboxylate (350 mg, 0.91 mmol) in THF (5 mL), H₂O (5 mL) and ⁿBuOH (5 mL) was added compound **4** (181 mg, 1.36 mmol), CuSO₄ (23 mg, 0.09 mmol) and sodium L-ascorbate (36 mg, 0.18 mmol). The mixture was stirred at 70 °C under N₂ for 16 hr. The resulting mixture was concentrated and purified by flash column chromatography on silica gel (methanol in dichloromethane = 20% v/v) to obtain compound **6** as solid (300 mg, yield 63.5%). LC-MS (ESI) m/z: 520[M+H]⁺.

## Step 4: Synthesis of 1-methyl-4-(1-(*trans*-4-(4-(trifluoromethyl)benzyloxy) pyrrolidin-3-yl)-1H-1,2,3-triazol-4-yl)pyridin-2(1H)-one (compound 7) (Scheme 8)

To the solution of compound **6** (280 mg, 0.54 mmol) in DCM (10 mL) was added TFA (2 mL). The mixture was stirred at room temperature for 2 hr and concentrated in vacuum, the residue was adjusted to pH~8 with NaHCO$_3$ solution and extracted with EtOAc (50 mL x 3), the combined organics were washed with brine (100 mL), dried over anhydrous Na$_2$SO$_4$, filtered and concentrated to leave the crude compound **7** as oil (200 mg, crude). LC-MS (ESI) m/z: 420[M+H]$^+$.

## Step 5: Synthesis of 4-(1-(*trans*-1-acryloyl-4-(4-(trifluoromethyl)benzyloxy) pyrrolidin-3-yl)-1H-1,2,3-triazol-4-yl)-1-methylpyridin-2(1H)-one (compound MYF-03-146) (Scheme 8)

To the solution of compound **7** (180 mg, 0.43 mmol) in THF (10 mL) was added acryloyl chloride (39 mg, 0.43 mmol) and Et$_3$N (87 mg, 0.86 mmol). The mixture was stirred at 0 °C under N$_2$ for 1 hour, and then concentrated and purified by prep-HPLC to obtain compound **MYF-03–146** as solid (215 mg, yield 95.5%). LC-MS (ESI) m/z: 474[M+H]$^+$. $^1$H-NMR (400 MHz, DMSO-$d_6$): δ (ppm) 8.89 (d, *J*=5.8 Hz, 1 H), 7.78 (d, *J*=7.1 Hz, 1 H), 7.70 (d, *J*=8.2 Hz, 2 H), 7.53 (d, *J*=8.1 Hz, 2 H), 6.85 (s, 1 H), 6.76–6.53 (m, 2 H), 6.19 (dd, *J*=16.8, 2.3 Hz, 1 H), 5.73 (ddd, *J*=10.2, 5.1, 2.3 Hz, 1 H), 5.47 (d, *J*=25.4 Hz, 1 H), 4.75 (s, 2 H), 4.57 (d, *J*=24.5 Hz, 1 H), 4.31–3.57 (m, 4 H), 3.44 (s, 3 H).

## MYF-03-135

## Step 1: Synthesis of *trans*-tert-butyl 3-(4-(tetrahydro-2H-pyran-4-yl)-1H-1,2,3-triazol-1-yl)-4-(4-(trifluoromethyl)benzyloxy)pyrrolidine-1-carboxylate (compound 3) (Scheme 9)

**Scheme 9.** Synthetic method of MYF-03-135.

To the solution of compound **2** (300 mg, 0.78 mmol) in THF (10 mL), H$_2$O (10 mL) and $^n$BuOH (10 mL) was added 4-ethynyltetrahydro-2H-pyran (128 mg, 1.16 mmol), CuSO$_4$ (20 mg, 0.078 mmol) and sodium L-ascorbate (31 mg, 0.156 mmol). The mixture was stirred at 70 °C under N$_2$ for 16 hours. The resulting mixture was concentrated and purified by flash column chromatography on silica gel (Methanol in dichloromethane = 20% v/v) to obtain compound **3** as solid (300 mg, yield 77.9%). LC-MS (ESI) m/z: 497 [M+H]$^+$.

## Step 2: Synthesis of 4-(tetrahydro-2H-pyran-4-yl)-1-(*trans*-4-(4-(trifluoromethyl)benzyloxy)pyrrolidin-3-yl)-1H-1,2,3-triazole (compound 4) (Scheme 9)

To the solution of compound **3** (280 mg, 0.56 mmol) in DCM (10 mL) was added TFA (2 mL). The mixture was stirred at room temperature for 2 hr and concentrated in vacuum, the residue was adjusted to pH~8 with NaHCO$_3$ solution and extracted with EtOAc (50 mL x 3), the combined organics were washed with brine (100 mL), dried over anhydrous Na$_2$SO$_4$, filtered and concentrated to leave the crude compound **4** as oil (250 mg, crude). LC-MS (ESI) m/z: 397 [M+H]$^+$.

## Step 3: Synthesis of 1-(*trans*-3-(4-(tetrahydro-2H-pyran-4-yl)–1 H-1,2,3-triazol-1-yl)–4-(4-(trifluoromethyl)benzyloxy)pyrrolidin-1-yl)prop-2-en-1-one (compound MYF-03–135) (Scheme 9)

To the solution of compound **4** (230 mg, 0.58 mmol) in THF (10 mL) was added acryloyl chloride (52 mg, 0.58 mmol) and Et$_3$N (117 mg, 1.16 mmol). The mixture was stirred at 0 °C under N$_2$ for

1 hour, and then concentrated and purified by prep-HPLC to obtain compound **MYF-03–135** as solid (180 mg, yield 69.0%). LC-MS (ESI) m/z: 451[M+H]⁺. ¹H-NMR (400 MHz, DMSO-$d_6$): δ (ppm) 8.05 (d, J=6.6 Hz, 1 H), 7.70 (d, J=7.6 Hz, 2 H), 7.51 (d, J=7.8 Hz, 2 H), 6.66–6.57 (m, 1 H), 6.18 (d, J=16.8 Hz, 1 H), 5.77–5.67 (m, 1 H), 5.43–5.29 (m, 1 H), 4.72 (d, J=5.2 Hz, 2 H), 4.50 (dd, J=31.4, 5.0 Hz, 1 H), 4.24–3.96 (m, 2 H), 3.89 (d, J=13.1 Hz, 2 H), 3.82 (td, J=13.1, 4.4 Hz, 1 H), 3.58 (dd, J=13.1, 3.3 Hz, 1 H), 3.44 (t, J=11.3 Hz, 2 H), 2.93 (t, J=10.6 Hz, 1 H), 1.85 (d, J=12.9 Hz, 2 H), 1.61 (q, J=11.4 Hz, 2 H).

## MYF-03-176

### Step 1: Synthesis of (3R,4R)-tert-butyl 3-azido-4-hydroxypyrrolidine-1-carboxylate (compound 3) (Scheme 10)

**Scheme 10.** Synthetic method of MYF-03-176.

A mixture of compound **1** (4000 mg, 21.6 mmol), compound **2** (2664 mg, 23.2 mmol) and catalyst **3** (328 mg, 0.42 mmol) was stirred at rt for overnight under N₂ protection. The reaction mixture was treated with MeOH (60 mL) and K₂CO₃ (1788 mg, 12.8 mmol) and continued to stirring for 5 hr. The reaction mixture was extracted with ethyl acetate (300 mL x 3), and washed by water (300 mL x 2). The organic layer was dried over Na₂SO₄ and concentrated. The residue was purified by flash column chromatography on silica gel (ethyl acetate in petroleum ether = 20% v/v) to obtain 3.5 g product compound **3** as yellow oil (3.5 g, yield 71%). LC-MS (ESI) m/z: 129[M+H]⁺.

### Step 2: Synthesis of (3R,4R)-tert-butyl 3-azido-4-(4-(trifluoromethyl)benzyloxy)pyrrolidine-1-carboxylate (compound 5) (Scheme 10)

A mixture of compound **3** (500mg, 2.19 mmol), compound **4** (524mg, 2.19 mmol) and NaH (105 mg, 2.62 mmol) in THF (10 mL) was stirred at rt for 6 h under N₂ protection. The reaction mixture was monitored by LC-MS. The reaction mixture was extracted with ethyl acetate (100 mL), and washed by water (50 mL). The organic layer was dried over Na₂SO₄, was concentrated and purified by column chromatography on silica gel (ethyl acetate in petroleum ether = 30% v/v) to obtain 600 mg product compound **5** as white oil (600 mg, yield 95%). LC-MS (ESI) m/z: 287[M+H]⁺.

### Step 3: Synthesis of (3R,4R)-tert-butyl 3-amino-4-(4-(trifluoromethyl)benzyloxy)pyrrolidine-1-carboxylate (compound 7) (Scheme 10)

A mixture of compound **5** (1000mg, 2.58 mmol), PPh₃ (814 mg, 3.1 mmol) and H₂O (930 mg, 51.6 mmol) in THF (40 mL) was stirred at 70 °C for 5 h under N₂ protection. The reaction mixture was monitored by LC-MS. The reaction mixture was extracted with ethyl acetate (300 mL), and washed by water (200 mL), The organic layer was dried over Na₂SO₄, concentrated and purified by p-HPLC to obtain 800 mg product compound **7** as yellow oil (800 mg, yield 86%). LC-MS (ESI) m/z: 261[M+H-100]⁺.

### Step 4: Synthesis of (3R,4R)-tert-butyl 3-(pyrimidin-2-ylamino)-4-(4-(trifluoro-methyl)benzyloxy)pyrrolidine-1-carboxylate (compound 9) (Scheme 10)

A mixture of compound **7** (600 mg, 1.6 mmol), compound **8** (240 mg, 1.84 mmol) and DIPEA (420 mg, 3.24 mmol) in *n*BuOH (6 mL) was stirred at 70 °C for overnight under $N_2$ protection. The reaction mixture was monitored by LC-MS. The reaction mixture was concentrated and purified by p-HPLC to obtain 500 mg product compound **9** as white oil (500 mg, yield 71%). LC-MS (ESI) m/z: 439[M+H]$^+$.

### Step 5: Synthesis of N-((3R,4R)-4-(4-(trifluoromethyl)benzyloxy)pyrrolidin-3-yl)pyrimidin-2-amine (compound 10) (Scheme 10)

A mixture of compound **9** (400mg, 0.91 mmol) and TFA (1 mL) in DCM (3 mL) was stirred at rt for 2 hr under $N_2$ protection. The mixture was concentrated to obtain 300 mg crude of compound **10** as yellow oil. Used for next step (400 mg, yield 97%). LC-MS (ESI) m/z: 339[M+H]$^+$.

### Step 6: Synthesis of 2-fluoro-1-((3R,4R)-3-(pyrimidin-2-ylamino)-4-(4-(tri-fluoromethyl)benzyloxy)pyrrolidin-1-yl)prop-2-en-1-one (MYF-03-176) (Scheme 10)

A mixture of compound **10** (200mg, 0.58 mmol), compound **11** (60mg, 0.69 mmol) and HATU (256 mg, 0.69 mmol) and DIEA (224 mg, 1.74 mmol) in DMF (5 mL) was stirred rt for overnight under $N_2$ protection. The reaction mixture was monitored by LC-MS. The reaction mixture was concentrated and purified by p-HPLC to obtain 150 mg product **MYF-03–176** as white solid (150 mg, yield 63%). LC-MS (ESI) m/z: 411[M+H]$^+$. 1 H NMR (400 MHz, MeOD) δ 8.44 (s, 2 H), 7.82–7.50 (m, 4 H), 6.83 (dd, J=8.3, 4.9 Hz, 1 H), 5.50 (dt, J=47.1, 3.3 Hz, 1 H), 5.27 (dt, J=16.5, 3.4 Hz, 1 H), 4.82 (dd, J=13.3, 8.8 Hz, 2 H), 4.68–4.57 (m, 1 H), 4.28–3.67 (m, 5 H). $^{13}$C NMR (126 MHz, DMSO-$d_6$) δ 161.47 (d, *J*=6.9 Hz), 159.38 (t, *J*=32.4 Hz), 158.07, 157.84, 155.72, 143.25, 128.18, 127.96 (d, *J*=9.0 Hz), 125.41, 125.16 (q, *J*=3.7 Hz), 123.25, 110.90 (d, *J*=5.0 Hz), 100.03 (dd, *J*=18.7, 15.5 Hz), 80.99 (d, *J*=3.2 Hz), 78.42, 69.28, 55.19 (d, *J*=2.9 Hz), 52.45, 51.21 (d, *J*=9.0 Hz), 50.85 (d, *J*=9.0 Hz), 50.60, 50.23. HRMS (ESI) calcd. for C19H19F4N4O2 (M+H)$^+$: 411.1439, Found: 411.1444.

## Acknowledgements

The authors thank the following for valuable help with this study: Dr. Milka. Kostic for her advice and editing for this manuscript; Jim Sun at the NMR facility of the Dana-Farber Cancer Institute for his assistance on $^1$H NMR data collection; Zachary Herbert from the Molecular Biology Core Facility at the Dana-Farber Cancer Institute for the sequencing services; Kara Soroko and Jessica Sarro from Dana-Farber Cancer Institute for the animal study services. JDM acknowledges this work was supported by the Hale Family Center for Pancreatic Cancer Research.

## Additional information

### Competing interests

Mengyang Fan, Yang Gao, Yao Liu: is one of the inventors on TEAD inhibitor patents (WO2020081572A1). Jianwei Che: is a consultant to Soltego, Jengu, Allorion, EoCys, and equity holder for Soltego, Allorion, EoCys, and M3 bioinformatics & technology Inc. Tinghu Zhang: is a scientific funder, equity holder and consultant in Matchpoint. T.Z. is one of the inventors on TEAD inhibitor patents (WO2020081572A1). Nathanael S Gray: is a founder, science advisory board (SAB) member and equity holder in Syros, Jengu, C4, B2S, Allorion, Inception, GSK, Larkspur (board member), Soltego (board member) and Matchpoint. The Gray lab receives or has received research funding from Novartis, Takeda, Astellas, Taiho, Janssen, Kinogen, Voronoi, Interline, Springworks and Sanofi. TEAD inhibitors developed in this manuscript are licensed to a start-up (Lighthorse) where Gray has a financial interest. N.S.G. is one of the inventors on TEAD inhibitor patents (WO2020081572A1). The other authors declare that no competing interests exist.

## Funding

| Funder | Grant reference number | Author |
|---|---|---|
| Epiphanes | | Tinghu Zhang<br>Nathanael S Gray |
| Hale Family Center for Pancreatic Cancer Research | | Joseph D Mancias |

The Gray lab has sponsored research agreement for TEAD inhibitor project with Epiphanes. The funders had no role in study design, data collection and interpretation, or the decision to submit the work for publication.

## Author contributions

Mengyang Fan, Conceptualization, Investigation, Writing – original draft, Project administration, Writing – review and editing; Wenchao Lu, Data curation, Validation, Investigation, Writing – review and editing; Jianwei Che, Miljan Kuljanin, Data curation, Formal analysis, Investigation, Writing – review and editing; Nicholas P Kwiatkowski, Supervision; Yang Gao, Yao Liu, Ezekiel A Geffken, Jimit Lakhani, Kijun Song, Jason Tse, Investigation; Hyuk-Soo Seo, Formal analysis, Investigation, Visualization, Writing – review and editing; Scott B Ficarro, Data curation, Formal analysis, Investigation, Visualization, Methodology, Writing – review and editing; Prafulla C Gokhale, Data curation, Formal analysis, Writing – original draft, Writing – review and editing; Wenzhi Ji, Jie Jiang, Zhixiang He, Project administration; Andrew S Boghossian, Matthew G Rees, Melissa M Ronan, Jennifer A Roth, Data curation, Formal analysis, Investigation; Joseph D Mancias, Resources, Supervision, Investigation, Writing – review and editing; Jarrod A Marto, Resources, Supervision, Writing – review and editing; Sirano Dhe-Paganon, Resources, Data curation, Formal analysis, Supervision, Investigation, Visualization, Writing – review and editing; Tinghu Zhang, Conceptualization, Supervision, Funding acquisition, Investigation, Writing – original draft, Project administration, Writing – review and editing; Nathanael S Gray, Conceptualization, Resources, Supervision, Funding acquisition, Project administration, Writing – review and editing

## Author ORCIDs

Mengyang Fan ⓘ http://orcid.org/0000-0002-0871-8792
Wenchao Lu ⓘ http://orcid.org/0000-0003-1175-365X
Kijun Song ⓘ http://orcid.org/0000-0002-6037-9345
Jie Jiang ⓘ http://orcid.org/0000-0003-3795-672X
Andrew S Boghossian ⓘ http://orcid.org/0000-0002-7008-8138
Tinghu Zhang ⓘ http://orcid.org/0000-0003-1028-8020
Nathanael S Gray ⓘ http://orcid.org/0000-0001-5354-7403

## Ethics

Animals acclimated for at least 5 days before initiation of the study. All in vivo studies were conducted at Dana-Farber Cancer Institute with the approval of the Institutional Animal Care and Use Committee in an AAALAC accredited vivarium.

## Decision letter and Author response

Decision letter https://doi.org/10.7554/eLife.78810.sa1
Author response https://doi.org/10.7554/eLife.78810.sa2

# Additional files

## Supplementary files

• Supplementary file 1. Table of diffraction data collection and refinement statistics for TEAD1 and MYF-03–69 co-crystal structure.

• Supplementary file 2. Proteome-wide selectivity profile of MYF-03–69 on cysteines labeling using SLC-ABPP approach.

• Supplementary file 3. List of differentially expressed genes under MYF-03–69 treatments.

• Supplementary file 4. Area under the curve (AUC) data of PRISM cell viability screen and corresponding CERES scores of YAP1 and TEADs.

• Supplementary file 5. Correlation analysis of MYF-03–69 PRISM sensitivity profile and gene DepMap dependency scores.

• Supplementary file 6. Liver microsome stability and hepatocyte stability of MYF-03–176 and K-975.

• Supplementary file 7. Liver microsome stability and PK parameters of MYF-01–37, MYF-03–69 and MYF-03–176.

• MDAR checklist

## Data availability

Diffraction data have been deposited in PDB under the accession code 7LI5. RNA sequencing data have been deposited in BioSample database under accession codes SAMN19288936, SAMN19288937, SAMN19288938, SAMN19288939, SAMN19288940, SAMN19288941, SAMN19288942, SAMN19288943, SAMN19288944, SAMN19288945 and SAMN19288946. All datasets generated or analyzed during this study have been deposited in Dryad. Uncropped gels or blots image of Figure 1e, 2e, 3a, 3b, 4d and their related figure supplements 3, 4, 7 are provided as source data.

The following datasets were generated:

| Author(s) | Year | Dataset title | Dataset URL | Database and Identifier |
|---|---|---|---|---|
| Fan M | 2022 | Data from: Covalent disruptor of YAP-TEAD association suppresses defective Hippo signaling | https://doi.org/10.5061/dryad.rxwdbrvbn | Dryad Digital Repository, 10.5061/dryad.rxwdbrvbn |
| Seo H-S, Dhe-Paganon S | 2022 | Crystal Structure Analysis of human TEAD1 | https://www.rcsb.org/structure/7LI5 | RCSB Protein Data Bank, 7LI5 |

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
