## [Editor Report]

The Hippo signaling pathway has emerged as a key signaling pathway in cancer and many other diseases, but there is a lack of high-quality chemical tools that would enable functional studies. Fan et al. disclose the development of covalent TEAD inhibitors and report on the therapeutic potential of this class of agents in the treatment of TEAD-YAP-driven cancers. The authors' claims and conclusions are well supported by the data. The optimized derivative represents a clear improvement from previously reported compounds and provides a high-quality probe for future studies.

---

## [Decision Letter]

**Decision letter after peer review:**

Thank you for submitting your article "Covalent Disruptor of YAP-TEAD Association Suppresses Defective Hippo Signaling" for consideration by *eLife*. Your article has been reviewed by 2 peer reviewers, and the evaluation has been overseen by a Reviewing Editor and Kevin Struhl as the Senior Editor. The following individual involved in the review of your submission has agreed to reveal their identity: Stefan Knapp (Reviewer #1).

Essential revisions:

Fan et al. disclose the development of covalent TEAD inhibitors and report on the therapeutic potential of this class of agents in the treatment of TEAD-YAP-driven cancers. While reviewers recognize the impressive characterization of the new compound, there also expressed concerns about how the new compound differentiates from the previously reported (and closely related) compound K-975. In order for the study to represent a significant enough advance for *eLife*, the authors need to articulate how the new compound MYF-03-176 is better than K-975, and to demonstrate that occupation of the newly identified side pocket contributes to improved potency and/or specificity over known inhibitors.

*Reviewer #1 (Recommendations for the authors):*

This is an interesting report on a new chemical probe as well as its functional characterization in mesothelioma. MYF-03-69 is not the first TEAD inhibitor but it is comprehensively characterized enabling its use for functional studies. The data are conclusive and contain necessary controls. Given the long interest of the Gray lab in developing chemical tools, the level of characterization of MYF-03-69 is outstanding. I have only a few comments:

The characterized chemical probe, MYF-03-69 was unfortunately not suitable for in vivo use and a PK optimized inhibitor (MYF-03-176) was used. Even though this inhibitor is structurally highly similar, it has not been extensively characterized (only by proliferation assays and luciferase-based gene expression). It would be interesting to demonstrate if either selectivity, on target potency, or effects on endogenous gene expression of both inhibitors is similar.

At a high dose, there seems to be a significant weight loss until day 30. Later the mice seem to recover gaining weight again comparable to the control group. Do the authors have an explanation for this behavior?

*Reviewer #2 (Recommendations for the authors):*

1. This manuscript discloses the optimization and characterization of a previously reported compound (MYF-01-37, Kurppa et al. 2020 Cancer Cell). In comparison to MYF-01-37, improvements in biochemical- and cell-based potency, as well as PK properties, are clear. The impact on proapoptotic BMF gene expression and its possible use as a pharmacodynamic biomarker is intriguing. Are there additional new biological insights for this compound series or can the BMF findings be expanded?

2. Differentiation from what has been reported for K-975 (Keneda et al. 2020 Am J Cancer Res 10:4399. PMID 33415007), a very closely related structure (note the impact of para- versus meta-substitution with EWG of distal aryl ring system on potency for K-975 and MYF-03-69) is not clear. MYF-03-176 is equipotent when compared head-to-head with K-975 (IC50 = 11 nM for both compounds). The efficacious dose of MYF-03-176 that is reported here is equivalent to that reported for K-975 (30 mg/kg reported for both compounds). Is it possible to more clearly articulate differentiation from K-975, as well as new biological insights for this compound series and/or the general therapeutic approach? Is BMF a useful biomarker for this therapeutic class and is MYF-03-176 differentiated from K-975 with respect to in vivo activity?

3. Demonstration of on-target activity in vivo is not provided. Impact of MYF-03-176 treatment on tumor BMF levels and establishment of correlated minimal efficacious dose and associated Cmax exposure levels would significantly increase the ability to interpret evidence for on-target TEAD activity.

---

## [Author Response]

Essential revisions:Fan et al. disclose the development of covalent TEAD inhibitors and report on the therapeutic potential of this class of agents in the treatment of TEAD-YAP-driven cancers. While reviewers recognize the impressive characterization of the new compound, there also expressed concerns about how the new compound differentiates from the previously reported (and closely related) compound K-975. In order for the study to represent a significant enough advance for eLife, the authors need to articulate how the new compound MYF-03-176 is better than K-975, and to demonstrate that occupation of the newly identified side pocket contributes to improved potency and/or specificity over known inhibitors.

We appreciate the reviewer’s question.

The superiority of compound MYF-03-176 relative to K-975 comes from four main aspects:

1. Improved physicochemical and pharmacokinetic properties. It is well documented that as a covalent warhead, an aliphatic acrylamide, such as the one we employed in MYF-03-176, is less electrophilic than an aromatic acrylamide (the one found in K-975), and, therefore, more resistant to GSH conjugation and more metabolically stable (J. Med. Chem. 2014, 57, 10072–10079; Eur. J. Med. Chem. 2018, 160, 94–107). Furthermore, to improve pharmacokinetic properties even further, we introduced a fluoride atom to the acrylamide warhead, which further reduced the risk of GSH conjugation as has also recently been reported for the covalent G12C KRAS inhibitor MRTX series (J. Med. Chem. 2020, 63, 6679–6693). Our mouse liver microsome stability and hepatocyte stability results confirm the improved stability of MYF-03-176 over K-975. As evidenced below, K-975 has an extremely short half-life. Across the board, MYF-03-176 exhibits more favorable physicochemical and pharmacokinetic properties, suggesting that MYF-03-176 is a superior candidate for further drug discovery and development, and an improved pharmacological probe compound. We have included this data in the revised manuscript (Supplementary File 6).

2. Improved selectivity. We present evidence that MYF-03-69 occupies a new pocket, which provides a better chance to incorporate additional specificity features in addition to determinants of binding to the central palmitate binding pocket. MYF-03-69 design takes advantage of this opportunity and the molecule forms much more extensive contacts with the target (see Figure 1—figure supplement 3 for the superimposition of cocrystal structures of MYF-03-69 [magenta] and K-975 [green] in TEAD1 [PDB code: 7LI5 and 7CMM], respectively). K-975 only occupies the palmitate pocket, which explains why K-975 needs to primarily be hydrophobic. In comparison, MYF-03-69 has a polar group which extends into the side pocket which improves its ‘drug-like’ property and improves its selectivity. We now stress this point more strongly in the text.

3. Decreased off-target effects. Given the relatively small size, the potency of K-975 might be driven by the high reactivity of its electrophilic warhead, which raises concerns about promiscurity and non-specific cell toxicity. While there is no standard chemo-protemic protocal to evaluate the selectivity of cross labeling on other protein targets, we tested antiproliferative effect of MYF-03-176 and K-975 in NCI-H2452 (mesothelioma cells with intact Hippo signaling). We found that K-975 inhibited cell proliferation much more potently than MY-03-176 (at 30 mM), which suggests that K-975 may have non-specific cell toxicity at high concentration probably due to its reactive covalent warhead.

**Author response image 1. sa2fig1:** 

4. Improved in vivo efficacy. In terms of in vivo efficacy, when NCI-H226 cells derived xenograft mice were dosed with 30 mg/kg K-975 twice daily, the tumors kept growing and reach more than 1.5-fold volume on 14^th^ day. While with the same dosage, MYF-03-176 showed a significant tumor regression. K-975 did not reach such efficacy even with 100 or 300 mg/kg twice daily, either in NCI-H226 or MSTO-211H CDX mouse model according to the paper (Keneda et al. 2020 Am J Cancer Res 10:4399).Therefore, due to the improved physicochemical and pharmacokinetic properties, as well as improved specificity, decreased off-target effects, and improved efficacy MYF-03-176 represents a significant advancement over K-975. We expect that MYF-03-176 will become a pharmacological tool compound of choice for studying TEAD1 biology, as well as offer opportunities for further drug development.

Reviewer #1 (Recommendations for the authors):This is an interesting report on a new chemical probe as well as its functional characterization in mesothelioma. MYF-03-69 is not the first TEAD inhibitor but it is comprehensively characterized enabling its use for functional studies. The data are conclusive and contain necessary controls. Given the long interest of the Gray lab in developing chemical tools, the level of characterization of MYF-03-69 is outstanding. I have only a few comments:

We thank the reviewer for kind and supportive comments!

The characterized chemical probe, MYF-03-69 was unfortunately not suitable for in vivo use and a PK optimized inhibitor (MYF-03-176) was used. Even though this inhibitor is structurally highly similar, it has not been extensively characterized (only by proliferation assays and luciferase-based gene expression). It would be interesting to demonstrate if either selectivity, on target potency, or effects on endogenous gene expression of both inhibitors is similar.

This is an excellent point and we provided this information in the Supporting Information (Figure 6—figure supplement 3). We repeated the experiment to compare the effects of K-975, MYF-03-69 and MYF-03-176 on the expression of of TEAD targets genes *CTGF*, *CYR61*, and *BMF* in NCI-H226 cells following compound treatment for 24 h. MYF-03-176 affects expression of these genes in the same way as MYF-03-69 does, but is more potent. As comparison in Author response image 2, we have included K-975 in this assay and clearly, it’s less potent compared to MYF-03-176.

In addition, we have established a biochemical assay to charaterize the analogs of MYF-03-69 and we are submitting a comprehensive medicinal chemistry publication detailing the SAR (structure-and activity relationship) for this series of compounds to a more specialized journal.

At a high dose, there seems to be a significant weight loss until day 30. Later the mice seem to recover gaining weight again comparable to the control group. Do the authors have an explanation for this behavior?

We thank the reviewer for this comment. We currently don’t know what caused this behavior, and we can only speculate that this may be due to renal toxicity, given that this effect has been reported in the work describing K-975. In that work, the authors observed proteinuria in rats and monkeys after a two-week treatment, which they linked to potential renal toxicity. Going forward, renal toxicity should be carefully monitored as it might be a likely liability for these compounds.

Reviewer #2 (Recommendations for the authors):1. This manuscript discloses the optimization and characterization of a previously reported compound (MYF-01-37, Kurppa et al. 2020 Cancer Cell). In comparison to MYF-01-37, improvements in biochemical- and cell-based potency, as well as PK properties, are clear. The impact on proapoptotic BMF gene expression and its possible use as a pharmacodynamic biomarker is intriguing. Are there additional new biological insights for this compound series or can the BMF findings be expanded?

We thank the reviewer for raising this point which we agree is worthy of further investigation. Upregulation of *BMF* could be a common observation in cells that are sensitive to TEADi beyond mesothelioma such as in liver cancer SKHEP-1 and 94T778. We also observed *BMF* upregulation with other reported TEAD inhibitors (for example, K-975 also led to upregulation of *BMF*), thus it may be useful, along with downregulation of *CTGF* and *CYR61,* as an indication of TEADi on-target activity. Our hope is that MYF-03-176, due to its improved properties over the existing compounds in this space, will become a ‘tool’ compound for the community, thus enabling us and others to discover novel aspects of TEAD- related biology.

2. Differentiation from what has been reported for K-975 (Keneda et al. 2020 Am J Cancer Res 10:4399. PMID 33415007), a very closely related structure (note the impact of para- versus meta-substitution with EWG of distal aryl ring system on potency for K-975 and MYF-03-69) is not clear. MYF-03-176 is equipotent when compared head-to-head with K-975 (IC50 = 11 nM for both compounds). The efficacious dose of MYF-03-176 that is reported here is equivalent to that reported for K-975 (30 mg/kg reported for both compounds). Is it possible to more clearly articulate differentiation from K-975, as well as new biological insights for this compound series and/or the general therapeutic approach? Is BMF a useful biomarker for this therapeutic class and is MYF-03-176 differentiated from K-975 with respect to in vivo activity?

Please see our detailed response above that articulates the improved properties of MYF-03-176 relative to K-975. In brief, MYF-03-176 has improved physicochemical and pharmacokinetic properties, as well as improved selectivity, decreased off-target toxicity and improved efficacy.

Regarding *BMF* gene expression as a biomarker for TEADi activity, we agree that the effect of *BMF* gene upregulation is worthy of further investigation. We observed *BMF* upregulation with other reported TEAD inhibitors (K-975, for example), thus *BMF* gene upregulation may be useful as a biomarker of TEAD inhibition, especially when used along with downregulation of *CTGF* and *CYR61*.

3. Demonstration of on-target activity in vivo is not provided. Impact of MYF-03-176 treatment on tumor BMF levels and establishment of correlated minimal efficacious dose and associated Cmax exposure levels would significantly increase the ability to interpret evidence for on-target TEAD activity.

We have performed pharmacodynamic studies and found that MYF-03-176 can downregulate downstream target genes including *CTGF*, *CYR61*, ANKRD1 and upregulate *BMF* as expected. We agree with the reviewer that developing a correlation between PK and a PD marker will be important to better understand whether drug exposure and target gene modulation correlate with efficacy. We are currently performing additional efficacy studies and plan to publish these findings in due course, but we felt there was value disclosing and making available MYF-03-176 as a tool compound for the research community given its improvement relative to what has been published to-date.

**Author response image 3. sa2fig3:**